# FAST SDP CERTIFICATION OF NEURAL NETWORKS : TOWARDS LARGE MULTI-CLASS DATASETS

## ABSTRACT

We present a new quadratic model for the certification problem in adversarial robustness, which simultaneously accounts for all possible target classes. Building on this model, we propose a novel semidefinite programming (SDP) relaxation for incomplete verification. A key advantage of our approach is that it certifies robustness in a single optimization, avoiding the need for a separate resolution per class. This yields a significant computational speed-up and enables scalability to large datasets with many classes. To further gain in efficiency, we also propose an effective pruning strategy of active neurons, thus reducing the problem dimensionality and accelerating convergence.

## 1 INTRODUCTION

Deep Neural Networks (DNNs) have achieved remarkable success and are widely implemented in various domains, including computer vision and natural language processing. This rapid adoption of DNNs has often prioritized efficiency and automation, overshadowing the crucial aspect of safety.

The research community has extensively studied various aspects of robustness, including out-of-distribution generalization, robustness to data corruption, and resistance to adversarial attacks. In particular, DNN have especially been proven vulnerable to adversarial attacks (Goodfellow et al., 2015), wherein malicious actors exploit the inherent complexity of these models to generate examples that deceive classifiers. This issue has raised concerns in many critical domains of applications of neural networks, like autonomous vehicles or robotics, where adversarial attacks could be a mean for malicious acts.

An *adversarial attack* consists of solving a constrained optimization problem to determine an adversarial example for a given data $x$, *i.e.*, a data in the neighborhood of $x$ which is classified differently by the DNN. These attacks represent a significant threat, particularly when the attacker has knowledge of the model architecture and parameters. In response, two main approaches have emerged to enhance the robustness of DNNs against such attacks: adversarial training and certified defenses. Adversarial training methods aim to improve robustness by performing adversarial augmentations. While these methods do offer increased resilience, they are not foolproof and can still be vulnerable to sophisticated attacks. On the other hand, certified defenses provide mathematical guarantees of robustness against adversarial attacks.

The certification problem for neural networks with ReLU activation functions is NP-complete (Katz et al., 2017). This inherent complexity implies that providing a *complete certification* requires substantial computational effort and remains limited in scalability. Many approaches solve combinatorial models to assess the DNNs predictions stability around each data. Several Mixed-Integer Programming (MIP) formulations were introduced to provide formal proofs of small ReLU DNNs (Tjeng et al., 2019; Fischetti & Jo, 2018; Cheng et al., 2017) but remain intractable for medium to large-scale problems.

Computing a non-negative lower bound is sufficient to certify that no adversarial attack is possible for a given target class. Thus, in order to speed up the certification, many approaches solve a relaxation of the original certification problem. In this paper, we focus on *incomplete* verifiers that provide lower bounds on the certification problem: a positive bound guarantees robustness while a negative bound is inconclusive. They constitute a compromise between efficiency and scalability, aiming to achieve the highest possible lower bound within a time limit. Most of the incomplete cer-

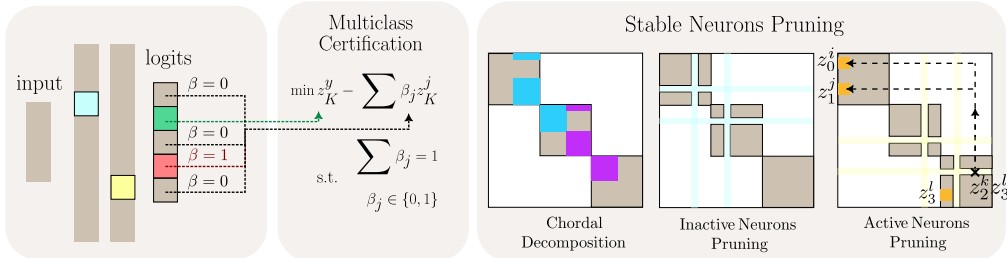

Figure 1: **Multiclass certification** The proposed method provides a new quadratic formulation for certifying a neural network across all labels simultaneously. This formulation relies on binary variables $\{\beta_j\}$ indicating the class associated with the worst adversarial example. To further reduce the number of variables, we also propose a pruning strategy that removes both inactive (blue) and active (yellow) neurons. Combined with a chordal decomposition of the SDP matrix (here represented by blue and purple constraints see Eq. (15)), this approach removes terms required to express the ReLU activations, which is compensated with the introduction of dedicated constraints see Eqs. (28) to (31). For instance, once neuron $z_2^k$ is removed, the quadratic interaction $z_2^k z_3^l$ is no longer represented in the matrix. To address this, we bound the dependencies of $z_3^l$ with respect to neurons from previous layers *ie.* $z_0^i$ and $z_1^j$.

tification methods are based on quadratic optimization formulations in which the ReLU is expressed as a quadratic non-convex equality from which a linear relaxation is computed (Wong & Kolter, 2018). Despite bringing promising results for certification, current approaches using SDP relaxations offer limited scalability in particular when certifying mid-to-large scale datasets composed of multiple classes. Indeed, these approaches are targeted ones, i.e. each combinatorial model tests if there exists an adversarial attack for one data and one *target* class. Thus, formally certifying a single data point requires looping over all possible target classes. This requirement can be rapidly cumbersome as modern datasets such as ImageNet-1k or ImageNet-21k propose hundreds or thousands of classes. Furthermore, as each neuron brings its own set of constraints in the optimization problem, current SDP approaches struggle for deep networks.

**Contributions** To deal with the aforementioned limitations, we introduce a new model for the certification problem that is based on an untargeted quadratic formulation $(QP_U)$ as illustrated in the left part of Fig. 1. This new approach allows us to certify each data by solving a *single* optimization problem and significantly speeds up the certification process. Our new formulation has the key advantage of preserving the non-negativity condition, certifying the data whenever a non-negative lower bound of $(QP_U)$ is obtained. We further introduce valid quadratic inequalities that tighten the bound of the relaxed problem. Finally, to scale up the certification, we propose an efficient pruning strategy able to remove *all stable* neurons from the constraints as can be seen in the right part of Fig. 1. This allows us to reduce the solution time. An interesting result is that this pruning strategy is generic and can be applied to other SDP relaxations. Finally, we present computational results demonstrating the efficiency of our methods against state-of-the-art approaches.

## 2 RELATED WORKS

**Certification problem** Complete certification methods aim to provide definitive guarantees about the absence of adversarial examples within a given input region. Seminal works include approaches based on Mixed-Integer Linear Programming (MILP), which model $ReLU$ activations through integer constraints, enabling exact reasoning over the network's activations. The MILP formulations proposed by (Fischetti & Jo, 2018; Tjeng et al., 2019) demonstrate formal verification for small-sized $ReLU$ networks. Similarly, satisfiability modulo theories solvers have been employed to provide sound and complete verification (Ehlers, 2017; Katz et al., 2017). However, a key limitation of complete verifiers remains their limited scalability to deep networks or high-dimensional datasets.

**Incomplete verifiers** aim to compute tight lower bounds on neural network robustness, providing formal certification guarantees whenever these bounds are sufficiently strong. A positive bound confirms robustness, while a negative one remains inconclusive, making these methods an interesting

compromise between scalability and theoretical soundness. Incomplete verifiers are divided into a wide variety of approaches including convex relaxations via duality (Wong & Kolter, 2018; Gowal et al., 2019), linear bounding of $ReLU$ activations (Weng et al., 2018), or discretized input space exploration (Huang et al., 2017). However, most incomplete verifiers may yield conservative bounds, and even well-optimized linear relaxations can fail to produce tight lower bounds over the objective.

**Semi Definite Programming (SDP) relaxations** have emerged as a promising class of incomplete verifiers for neural network verification as they produce sharper lower bounds than traditional linear programming approaches. While SDP methods are computationally more intensive, foundational works (Raghunathan et al., 2018; Zhang, 2020; Dathathri et al., 2018; Chiu & Zhang; Lan et al., b) have demonstrated empirical tightness compared to LP relaxations. This was further advanced by integrating geometric constraints such as triangle relaxations (Batten et al., 2021) and Reformulation-Linearization Technique (RLT) cuts (Lan et al., 2022), which refine the feasible region for $ReLU$ -activated networks. However, the relaxation becomes looser with increasing depth, and solving SDPs for deep networks often results in scalability challenges. This phenomenon is exacerbated when dealing with multiple classes as one SDP relaxation needs to be computed *for each target class* to achieve certification.

## 3 Preliminaries

Deep Neural Networks (DNNs) considered in this work are non-linear functions that map the input set to a measurable label set. They are described as successive layers given by the composition of a linear and a non-linear transformation.

Each layer $k$ contains $n_k$ neurons, indexed by $\mathcal{J}_k = \{1, ..., n_k\}$. The output $z_{k+1} \in \mathbb{R}^{n_{k+1}}$ of every layer $k \in [K-2]$ (*i.e.* $\{0, ..., K-2\}$) is computed by a $ReLU$ activation function: $z_{k+1} = ReLU(W_{k+1}z_k + b_{k+1})$, where $b_k \in \mathbb{R}^{n_k}$, $W_k \in \mathbb{R}^{n_k \times n_{k-1}}$ are the learned parameters of the network. Given a finite labeled dataset $\mathcal{D} = \{x_i, y_i\}$, the predicted class is given by $y^* = \text{argmax}_{j \in \mathcal{J}_K} z_K^j$ for data $x = z_0$ where $z_K^j$ is the $j^{th}$ component of vector $z_K = W_K z_{K-1} + b_K$.

For a given $\epsilon > 0$, the certification task verifies that for each data $(x, y)$ and all $z_0 \in \mathcal{B}_\epsilon(x)$ (the $\infty$-norm balls of center $x$ and radius $\epsilon$) the DNN correctly predicts the class $y$. Defining $\bar{\mathcal{J}}_K = \mathcal{J}_K \backslash \{y\}$, the set of all possible targets for a given sample, $W_K^j$, the $j^{th}$ row of matrix $W_K$, and $\mathcal{D}^+$, the set of well-classified data and their labels, we formally define robustness as follows:

**Property 1** (Targeted Robustness). *For a data $(x, y) \in \mathcal{D}^+$, a target class $j \in \bar{\mathcal{J}}_K$ and $\epsilon > 0$, a neural network is $(\epsilon, j)-$robust in $x$, if*

$$\min_{z_0 \in \mathcal{B}_\epsilon(x)} z_K^y - z_K^j \geq 0$$

**Property 2** (Full robustness). *For $\epsilon > 0$, a neural network is $\epsilon-$robust if for all $(x, y) \in \mathcal{D}^+$*

$$\min_{j \in \bar{\mathcal{J}}_K} \min_{z_0 \in \mathcal{B}_\epsilon(x)} z_K^y - z_K^j \geq 0 \tag{1}$$

Our aim in this paper is to determine whether a DNN satisfies the positivity Property 2. More formally, considering a data $(x, y) \in \mathcal{D}^+$, we consider the following optimization problem (Cert) defined for all $j \in \mathcal{J}_K$ :

$$(\text{Cert}) \begin{cases} \min_{z_0} \left(W_K^y z_{K-1} + b_K^y\right) - \left(W_K^j z_{K-1} + b_K^j\right) & \\ \text{s.t. } z_{k+1} = \text{ReLU}(W_{k+1}z_k + b_{k+1}) & k \in [K-2] \tag{2} \\ x - \epsilon \leq z_0 \leq x + \epsilon & \tag{3} \end{cases}$$

where Constraints (2) fix the output of the hidden layers and Constraint (3) ensures that $x$ belongs to $\mathcal{B}_\epsilon(x)$. The objective is the difference between the logit of the true class $y$ and the target class $j$.

Solving (Cert) to global optimality is hard due to the non-convexity of Constraints (2). However, by denoting $v(\text{Cert})$ the optimal value of (Cert), Property 2 is reached when $v(\text{Cert}) \geq 0$, for all $j \in \mathcal{J}_K$ and $(x, y) \in \mathcal{D}^+$. Thus, it is sufficient to compute a non-negative lower bound of (Cert) for all $j \in \mathcal{J}_K$ to ensure full robustness.

A quadratic formulation of (Cert) was introduced in (Raghunathan et al., 2018), obtaining the following targeted formulation:

$$(QP_T^j) \begin{cases} \min_{z_0} \left(W_K^y z_{K-1} + b_K^y\right) - \left(W_K^j z_{K-1} + b_K^j\right) \\ \text{s.t.} \quad z_{k+1} \geq 0, \quad z_{k+1} \geq W_{k+1} z_k + b_{k+1}, \;\; \forall k \in [K-2], \qquad (4) \\ \qquad z_{k+1} \odot (z_{k+1} - W_{k+1} z_k - b_{k+1}) = 0, \;\; \forall k \in [K-2], \qquad (5) \\ \qquad z_k \odot z_k - (L_k + U_k) \odot z_k + U_k \odot L_k \leq 0, \;\; \forall k \in [K-1] \qquad (6) \end{cases}$$

where $L_k$ and $U_k$ are lower and upper bounds over the preactivation vector of layer $k$. Constraints (4) combined with Constraints (5) are equivalent to Constraints (2). Constraints (6) can be rewritten as $(U_k - z_k) \odot (z_k - L_k) \geq 0$, which enforces $L_k \leq z_k \leq U_k$ when $L_k \leq U_k$. For $k = 0$, this is a quadratic equivalent to Constraints (3). Note that there exist efficient methods to propagate bounds across the network starting from the bounds of the input layer (e.g., $L_0 = x - \epsilon$ and $U_0 = x + \epsilon$ with the $\infty$ norm) (Wang et al., 2021) giving bounds $L_k$ and $U_k$ on the preactivation vector for all layers $k$.

**Property 3** (Target-positivity property). *If the optimal value of the targeted quadratic formulation $v(QP_T^j)$ is non-negative, the DNN satisfies Property 1.*

Due to the non-convexity of Constraints (5) and (6), solving formulation $(QP_T^j)$ to global optimality is impractical even for small-sized DNNs. However, Property 3 ensures that the development of suitable relaxations can be sufficient to certify the robustness. In particular, using semi-definite relaxations for quadratic programming was widely studied (Anstreicher, 2009) (see Sec. D.1). Let $P = [1 \quad z][1 \quad z]^T$ be the matrix that collects all the linear and quadratic terms in $(QP_T^j)$. Then, the semi-definite relaxation of targeted problem $(QP_T^j)$ has the form:

$$(SDP_T^j - IP) \begin{cases} \min_P \; \left(W_K^y P[z_{K-1}] + b_K^y\right) - \left(W_K^j P[z_{K-1}] + b_K^j\right) \\ \text{s.t.} \quad P[z_{k+1}] \geq 0, \quad P[z_{k+1}] \geq W_{k+1} P[z_k] + b_{k+1}, \;\; \forall k \in [K-2] \qquad (7) \\ \qquad \text{diag}\left(P[z_{k+1} z_{k+1}^\top] - W_{k+1} P[z_k z_{k+1}^\top]\right) = b_{k+1} P[z_k], \;\; \forall k \in [K-2] \qquad (8) \\ \qquad \text{diag}\left(P[z_k z_k^\top]\right) - (L_k + U_k) P[z_k] + U_k \odot L_k \leq 0, \;\; \forall k \in [K-1] \qquad (9) \\ \qquad P \succeq 0, P[1] = 1 \qquad (10) \end{cases}$$

From its definition, each element of $P$ is related to a given term in (Cert) and the symbolic indexing $P[.]$ is used to index the vectors of elements of matrix $P$. Constraints (7)–(9) correspond to the linearization of Constraints (4)–(6) by the matrix variable $P$.

Note that matrix $P$ exhibits a block diagonal structure. Leveraging chordal decomposition techniques (Vandenberghe & Andersen, 2015), as outlined in (Batten et al., 2021), $P$ can be decomposed into multiple submatrix variables. Specifically, the decomposition yields $K - 1$ matrix variables, each associated to two consecutives layers $k$ and $k+1$: $P_k = [1 \quad z_k \quad z_{k+1}][1 \quad z_k \quad z_{k+1}]^\top$. This decomposition allows to deal with multiple modest-sized SDP matrices rather than $P$. This change is expressed in the previous constraints (7)–(9) by injecting these matrices as:

$$\begin{cases} P_k[z_{k+1}] \geq 0, \quad P_k[z_{k+1}] \geq W_{k+1} P_k[z_k] + b_{k+1}, \;\; \forall k \in [K-2] \qquad (11) \\ \text{diag}\left(P_k[z_{k+1} z_{k+1}^\top] - W_{k+1} P_k[z_k z_{k+1}^\top]\right) = b_{k+1} P_k[z_k], \;\; \forall k \in [K-2] \qquad (12) \\ \text{diag}\left(P_k[z_k z_k^\top]\right) - (L_k + U_k) P_k[z_k] + U_k \odot L_k \leq 0, \;\; \forall k \in [K-1] \qquad (13) \\ P_k[z_{k+1}] \leq A_{k+1} P_k[z_k] + B_{k+1}, \;\; \forall k \in [K-2] \qquad (14) \end{cases}$$

The triangular constraint (14) introduced in (Ehlers, 2017) tightens the upper bounds of a neuron $j$ of layer $k$ according to its activation status with $A_k = l_k \odot W_k$, $B_k = l_k \odot (b_k - L_k) + \text{ReLU}(L_k)$.

The variable $l_k = \frac{\text{ReLU}(U_k) - \text{ReLU}(L_k)}{U_k - L_k}$ indicates whether a neuron is stable active ($l_k = 1$), stable inactive ($l_k = 0$), or unstable ($0 < l_k < 1$). In the case of a *stable active* neuron, *i.e.* $L_k^j \geq 0$ then $z_k^j \leq W_k^j z_{k-1} + b_k^j$. If it is *stable inactive*, *i.e.* $U_k^j \leq 0$ then $z_k^j \leq 0$. In the *unstable* case, constraint (14) reduces to $z_k^j \leq \frac{U_k^j}{U_k^j - L_k^j}(W_k^j z_{k-1} + b_k^j) + \frac{U_k^j}{U_k^j - L_k^j}(b_k^j - L_k^j)$ (see Appendix B.3).

It has been further improved in (Lan et al., 2022) with the addition of RLT (Reinforcement Linearization Technique) cuts, giving the following $(SDP_T^j)$ problem:

$$(SDP_T^j) \begin{cases} \min \left( W_K^y P_{K-2}[z_{K-1}] + b_K^y \right) - \left( W_K^j P_{K-2}[z_{K-1}] + b_K^j \right) \\ \text{s.t. } (11) - (14) \\ \quad P_k[(1 \; z_{k+1})(1 \; z_{k+1})^\top] = P_{k+1}[(1 \; z_{k+1})(1 \; z_{k+1})^\top] \quad (15) \\ \quad RLT(p) \quad\quad\quad\quad\quad\quad\quad\quad\quad\quad\quad\quad\quad\quad\quad\quad\quad (16) \\ \quad P_k = [1 \quad z_k \quad z_{k+1}][1 \quad z_k \quad z_{k+1}]^\top \succeq 0, P_k[1] = 1, \quad (17) \end{cases}$$

where the minimization over $P$ has been omitted for clarity of notation. Constraints (15) ensure the coherence of the variables across consecutive matrices $P_k$ and $P_{k+1}$. For $L_k \le z_k \le U_k$ $\forall k \in [K-1]$, Constraints (16) are RLT cuts (Sherali & Adams, 1990). They result from the product of linear valid inequalities (triangular constraint, bound constraints) to obtain quadratic inequalities. A percentage $p$ of these RLT cuts is carefully chosen by a heuristic. We refer the reader to the supplementary material Sec. B.5 for more details on RLT cuts.

Using $(SDP_T^j)$ to certify a DNN requires solving one SDP for each data $(x, y) \in \mathcal{D}^+$ and each possible target ($j \in \bar{\mathcal{J}}_K$). This leads to two significant drawbacks. First, the certification process fails to scale with the number of classes. Second, since solving multiple SDP relaxations for **each** data point is computationally demanding, the number of cuts must be restricted, which in turn weakens the tightness of the resulting bounds on the objective. To answer these limitations, we now introduce a new *untargeted* model that certifies a sample for all targets by solving only one SDP, thus considerably reducing the certification burden.

## 4 METHOD

### 4.1 A NEW QUADRATIC MODEL FOR FULL CERTIFICATION

To avoid solving $|\bar{\mathcal{J}}_K|$ SDPs for each data $(x, y) \in \mathcal{X}^+$, we design a new model that directly checks Property 2. As shown in the left part of Fig. 1, we thus introduce binary variables $(\beta_j)_{j \in \bar{\mathcal{J}}_K}$ with $\beta_j$ equals to 1 if and only if the worst adversarial example is of class $j$. Thus, the left-hand-side of Equation (1) can be obtained by minimizing $z_K^y - \sum_{j \in \bar{\mathcal{J}}_K} \beta_j z_K^j$. Using this objective, we define the following quadratic formulation of the *full-robustness* (see Property 2) of a DNN:

$$(QP_U) \begin{cases} \min W_K^y z_{K-1} + b_K^y - \sum_{j \in \bar{\mathcal{J}}_K} \beta_j \left( W_K^j z_{K-1} + b_K^j \right) \\ \text{s.t. } (4) - (6) \\ \quad \sum_{j \in \bar{\mathcal{J}}_K} \beta_j = 1 \quad\quad\quad\quad\quad\quad\quad\quad\quad\quad\quad\quad\quad (18) \\ \quad \beta_j \in \{0, 1\} \quad\quad\quad\quad\quad\quad\quad\quad\quad\quad j \in \bar{\mathcal{J}}_K \quad (19) \end{cases}$$

Constraint (18) ensures that only one binary variable $\beta_j$ will be non-zero. We now prove that $v(QP_U)$ coincides with the lowest value of $v(QP_T^j)$ over all target classes.

**Theorem 1.** *Given a data $(x, y) \in \mathcal{D}^+$, and $\epsilon > 0$, we have $v(QP_U) = \min_{\bar{j} \in \bar{\mathcal{J}}_K} v(QP_T^{\bar{j}})$*

From this theorem, we can deduce that the non-negativity of $v(QP_U)$ ensures the full robustness. Indeed, an optimal value of $(QP_U)$ will set $\beta_{\bar{j}} = 1$ for the target class $\bar{j}$ which minimizes $z_K^y - z_K^{\bar{j}}$.

**Property 4** (Full positivity property). *If $v(QP_U)$ is non-negative, the DNN satisfies Property 2.*

$(QP_U)$ has only $|\bar{\mathcal{J}}_K|$ additional binary variables and 1 more constraint than $(QP_T^j)$. Note that one can prune some of these binary variables based on the lower and upper bounds of their logits: a class $j \in \bar{\mathcal{J}}_K$ can be pruned if there exists another class $\tilde{j} \in \bar{\mathcal{J}}_K$ such that $U_K^j \le L_K^{\tilde{j}}$ since the associated $\beta_j$ will be zero in any optimal solution of $(QP_U)$. Indeed, if there exists an adversarial attack, the

most damaging one would not target class $j$, but a class in $\bar{\mathcal{J}}_K \setminus \{j\}$. In the following, we assume that the dominated classes are excluded from the set $\bar{\mathcal{J}}_K$ B.1. This implies $\bigcap_{j \in \bar{\mathcal{J}}_K} [L_K^j, U_K^j] \neq \emptyset$.

Similarly to $(QP_T^j)$, the direct solution of $(QP_U)$ is impractical even for small-sized DNNs. Thus, we build a tight semi-definite relaxation of $(QP_U)$ that may certify the DNN using Property 4.

## 4.2 A TIGHT SDP RELAXATION

In order to handle the additional binary variables $\beta_j$ in our SDP relaxation, for $k \in [K-3]$, we use $P_k$ as defined in Constraint (17) and we introduce an additional matrix to linearize the products $\beta_j z_K$, i.e. $P_{K-2} = [1 \quad z_{K-2} \quad z_{K-1} \quad z_K \quad \beta][1 \quad z_{K-2} \quad z_{K-1} \quad z_K \quad \beta]^\top$. We build the following SDP relaxation of $(QP_U)$:

$$(SDP) \begin{cases} \min \ W_K^y \ P_{K-2}[z_{K-1}] + b_K^y - \sum_{j \in \bar{\mathcal{J}}_K} \left( W_K^j \ P_{K-2}[\beta_j z_K] + b_K^j \right) \\[2mm] \text{s.t. } (7) - (15) \\[1mm] \qquad \sum_{j \in \bar{\mathcal{J}}_K} P_{K-2}[\beta_j] = 1 \hfill (20) \\[2mm] \qquad \operatorname{diag}\left(P_{K-2}[\beta\beta^\top]\right) = P_{K-2}[\beta] \hfill (21) \\[1mm] \qquad P_k \succeq 0 \quad k \in [K-2] \hfill (22) \end{cases}$$

To tighten $(SDP)$, we use several cuts. First, to tighten the linearization of products $\beta$ by $z$, we use McCormick cuts (McCormick, 1976) for $k \in \{K-2, K-1\}$ present in matrix $P_{K-2}$, that are defined as follows:

$$\beta_j z_k \geq 0, \ \beta_j z_k \geq U_k \beta_j + z_k - \beta_j, \ \beta_j z_k \leq U_k \beta_j, \ \beta_j z_k \leq z_k \tag{23}$$

Similarly, we tighten the products $\beta_{j_1} z^{j_2}$ of the objective function by use of the McCormick envelopes for all $(j_1, j_2) \in \mathcal{J}_K \times \mathcal{J}_K$:

$$\beta_{j_1} z_K^{j_2} \leq z_K^{j_2} - L_{K-1}^{j_2}(\beta_{j_1} - 1), \ \beta_{j_1} z_K^{j_2} \leq U_K^{j_2} \beta_{j_1}, \ \beta_{j_1} z_K^{j_2} \geq z_K^{j_2} - U_K^{j_2}(\beta_{j_1} - 1), \ \beta_{j_1} z_K^{j_2} \geq L_K^{j_2} \beta_{j_1} \tag{24}$$

Then, we build 3 new specific families of valid quadratic cuts for $(QP_U)$. First, since $\beta$ is a unit vector, a subset of entries can be fixed, reducing the number of terms that require explicit modeling, and we get:

$$\beta_{j_1} \beta_{j_2} = 0 \quad \forall (j_1, j_2) \in \bar{\mathcal{J}}_K, j_1 \neq j_2 \tag{25}$$

Finally, for all pairs of two distinct adversarial targets $j_1, j_2$, we introduce new inequalities that leverage the specific structure of the certification problem. These constraints allow coupling the variables $\beta_{j_1}, \beta_{j_2}, z_K^{j_1}, z_K^{j_2}, \beta_{j_1} z_K^{j_1}$ and $\beta_{j_2} z_K^{j_2}$ using only two constraints. Intuitively, constraint 27 encourages the logit of the adversarial target selected by the model to exceed that of other possible targets. The two constraints are given below:

$$\begin{cases} \beta_{j_2} z_K^{j_2} \leq (1 - \beta_{j_1}) z_K^{j_1} + \beta_{j_2} U_K^{j_2} - (1 - \beta_{j_1}) L_K^{j_1} + \beta_{j_2}(L_K^{j_1} - z_K^{j_1}) & (26) \\[1mm] \beta_{j_2} z_K^{j_2} \geq z_K^{j_1} - (1 - \beta_{j_2}) U_K^{j_1} & (27) \end{cases}$$

**Proposition 1.** *Constraints (26) and (27) are valid inequalities of $(QP_U)$.*

## 4.3 PRUNING OF STABLE ACTIVE NEURONS

Formulating the certification problem with the smallest possible model is crucial to accelerating its resolution. With this in mind, recent works have shown (Jung et al., 2020) that a significant fraction of $ReLU$ units can become inactive across training and stay inactive under small perturbations. DNN can thus achieve natural sparsity after training, which can be leveraged to reduce the number of variables in the certification model *cf.* right part of Fig. 1. In modern architectures commonly used in computer vision, more than half of the neurons can be inactive (Kurtz et al., 2020; Tjeng et al., 2019), directly translating into an equivalent reduction in the number of variables. Beyond

inactive units, stable **active** neurons should also be taken into account (see (Serra et al.; Botoeva et al.)), thus achieving a far more compact formulation of the certification problem than in the original optimization problem.

The variable $z_k^j$ corresponding to a stable active neuron can be replaced by its linear expression $W_k^j z_{k-1} + b_k^j$. By recursively substituting each stable active neuron of vector $z_{k-1}$ by its linear expression in terms of $z_{k-2}$, and so on, the neuron $z_k^j$ can be linearly expressed across multiple layers.

The chordal decomposition that we consider in $(SDP_T^j)$ and $(SDP_U)$ uses matrices $P_k$ which only model the links between two consecutive layers. However, unmodeled quadratic terms appear in Constraint (5), i.e. $z_{k+1}^j(z_{k+1}^j - W_k z_k + b_k) = 0$. Indeed, let $z_k^a$ be the sub-vector of active neurons of layer $k$, and $z_k^u$ the sub-vector of unstable neurons (inactive neurons have been removed), Constraint (5) becomes $z_{k+1}^j(z_{k+1}^j - W_{k,u}^j z_k^u - b_k) = z_{k+1}^j W_{k,a}^j z_k^a = \sum_{l=0}^{k-1} A_l^i + B z_{k+1}^j$, where $A$ and $B$ are derived from products of linear layer weights (more information in Appendix B.2). This formulation leads to new cross-layer dependencies as products $z_{k+1}^j z_l^i$ between non consecutive layers $l$ and $k+1$ appear, which are not represented in matrices from the chordal decomposition. To keep this decomposition in our SDP relaxation, we bound the obtained right-hand side. As illustrated in the right part of Fig. 1, we use McCormick cuts based on the bounds of $z_{k+1}^j$ and $z_l^i$ to create two linear upper bounds and two linear lower bounds on the quadratic products $z_{k+1}^j z_l^i$ (for more information see B.2). We then substitute our quadratic terms by their linear bounds in the *ReLU* constraint and obtain the following four relaxed *ReLU* constraints :

$$\begin{cases} z_{k+1}^j(z_{k+1}^j - W_{k,u}^j z_k^u - b_k) \leq (C_1 + B)z_{k+1}^j & (28) \\[2mm] z_{k+1}^j(z_{k+1}^j - W_{k,u}^j z_k^u - b_k) \leq (C_2 + B)z_{k+1}^j & (29) \\[2mm] z_{k+1}^j(z_{k+1}^j - W_{k,u}^j z_k^u - b_k) \geq \sum_{l=0}^{k-1} C_{3,l}z_l + C_{3,k+1}z_{k+1}^j + C_3 & (30) \\[2mm] z_{k+1}^j(z_{k+1}^j - W_{k,u}^j z_k^u - b_k) \geq \sum_{l=0}^{k-1} C_{4,l}z_l + C_{4,k+1}z_{k+1}^j + C_4 & (31) \end{cases}$$

where the coefficients of $C_{k-1}$ are a linear combination of the coefficients $A_{k-1}^a$ and the lower and upper bounds on products of $z_k z_{k-1}$ defined by the McCormick envelopes.

We prune active neurons on all layers except the penultimate layer, which is in the objective function of $(SDP_T^j)$. Finally, we obtain the following enhanced SDP relaxation, where $P_k$ matrices have been truncated:

$$(SDP_U)\begin{cases} \min W_K^y P_{K-2}[z_{K-1}] + b_K^y - \sum_{j \in \bar{\mathcal{J}}_K}(W_K^j P_{K-2}[\beta_j z_{K-1}] + b_K^j) \\[2mm] \text{s.t. } (7)-(15),\ (17),\ (20)-(31) \end{cases}$$

Note that this pruning strategy of stable active neurons is a generic approach that can be applied to any SDP relaxation, either targeted or multiclass. As shown by our experiences of Section 5.3, applying this strategy to $(SDP_U)$ or $(SDP_T)$ clearly speeds up the resolution. This size reduction comes at the cost of relaxing some equality constraints with inequalities Eqs. (28) to (31). However, the new constraints added to our formulation counterbalance this relaxation, ensuring that the overall certification performances remain competitive.

By denoting $n_k^a$ the number of stable active neurons on layer $k$, and $n_k^u$ the number of unstable neurons, Proposition 2 specifies the reduction of size resulting from pruning.

**Proposition 2.** *The pruning of active neurons reduces the dimensions of each matrix variable $P_k$ for $k \in [K-3]$ from $(1 + n_k^a + n_k^u + n_{k+1}^a + n_{k+1}^u)^2$ to $(1 + n_k^u + n_{k+1}^u)^2$, and reduces $P_{K-2}$ in $SDP_U$ from $\left(1 + n_{K-2}^a + n_{K-2}^u + n_{K-1}^a + n_{K-1}^u)^2 + |\bar{\mathcal{J}}_K|\right)^2$ to $\left(1 + n_{K-2}^u + n_{K-1}^u)^2 + |\bar{\mathcal{J}}_K|\right)^2$.*

| Network | PGD | $SDP_U$ (ours) | | $SDP_T$ | | $SDP_{T,layer}$ | | $SDP_T$-IP | | $\beta$-CROWN | |
|---------|-----|------|------|------|------|------|------|------|------|------|------|
| | | Cert. | Time | Cert. | Time | Cert. | Time | Cert. | Time | Cert. | Time |
| 6x100 | 96 | **86** | 323 | 74 | 399 | 72 | 22 | 72 | 100 | 69 | 2 |
| 6x200 | 99 | **85** | 441 | 74 | 2109 | 67 | 87 | 69 | 2154 | 64 | 3 |
| 9x100 | 95 | **77** | 925 | 35 | 2614 | 26 | 212 | 27 | 1634 | 22 | 4 |
| 9x200 | 100 | **71** | 1679 | 53 | 4081 | 47 | 650 | 47 | 4483 | 43 | 5 |

Table 1: Comparison of our untargeted method $SDP_U$ with other SDP approaches from literature. Column PGD is an overestimation of actual robustness.

## 5 RESULTS

### 5.1 IMPLEMENTATION DETAILS

We ran our experiments on a Linux machine on a 64-core CPU and a 264Go RAM. We use the Python API of the MOSEK optimizer (MOSEK ApS, 2019), and fixed the number of threads to 4.

We evaluate $SDP_U$ (see details in Appendix B) on MNIST (10 classes). We reproduced the evaluation protocol from previous works on DNN certification (Raghunathan et al., 2018; Batten et al., 2021; Lan et al., 2022) by considering 4 different fully connected neural networks adversarially trained with PGD attacks (see Appendix Sec. D.3). Neural networks used are 6x100 and 9x100 from (Singh et al., 2019) tested under the same $\epsilon = 0.026$; 6x200 and 9x200 from (Singh et al., 2019) tested under the same $\epsilon = 0.015$. We have reproduced these networks to the best of our knowledge and report the detailed architecture and adversarial training in Appendix C.3 for future reproducibility. We conducted our first two experiments on 100 data points: the first 10 images of each class from the MNIST train set.

### 5.2 STATE-OF-THE-ART COMPARISON

To fully evaluate the proposed method with respect to other incomplete verifiers, we compare with the following methods: $\beta$-CROWN method from (Wang et al., 2021); $SDP_T$-IP from (Raghunathan et al., 2018), $SDP_{T,layer}$ from (Batten et al., 2021) with a chordal decomposition of matrices, ablation of stable inactive neurons, and the triangular constraint ; $SDP_T$ from (Lan et al., 2022) with a chordal decomposition of matrices, ablation of stable inactive neurons, the triangular constraint, and 10% of the RLT cuts ; $SDP_U$ with a chordal decomposition of matrices, ablation of stable inactive and active neurons, the triangular constraint, and 100% of the RLT cuts for 6x100 and 9x100, 60% for 6x200 and 9x200. The results are reported in Tab. 1, where each line corresponds to one network. We ran the experiments of targeted SDP models across all non trivially certified adversarial targets after inspection of the logit bounds (see the number of remaining targets in Table 3 of Appendix C.3 and the algorithms used in B.1). The bounds on the preactivation values are computed with $\alpha - \beta$-CROWN (Wang et al., 2021). Column PGD is an overestimation of actual robustness, and for each method Column *Cert.* is the percentage of full robustness (across *all* targets), and Column *Time* is the mean total runtime per image (seconds) across all classes.

$\beta$-CROWN is GPU-accelerated and offers fast verification; however, it is unable to tighten the bounds sufficiently to eliminate all possible target classes when the number of neurons increases, as in 9x100 and 9x200. Method $SDP_T$-IP does not involve chordal decomposition of matrix variables nor efficient cuts. As expected, it reaches a low certification average percentage within a huge computation time. The impact of the introduction of both pruning of inactive neurons and chordal decomposition of matrices on the computation time can be observed with methods $SDP_{T,layer}$, $SDP_T$, and $SDP_U$. For $SDP_{T,layer}$, we observe a clear speed-up of the certification process (in comparison with $SDP_T$-IP), but with a very low certification percentage (even sometimes lower than $SDP_T$-IP). The use of RLT cuts in $SDP_T$ improves the certification percentage, but slows the full computation. Note that since $SDP_T$ needs to solve up to 9 SDP models to certify, the cost of adding the RLT cuts is significant, and only a small proportion can be added (10% in our experiments) to keep tractability. Finally, we observe that our new method $SDP_U$ achieved the best performances by a clear margin in terms of certification percentage (increased by 19 percentage points on average). Furthermore, it achieves the best certification and computation time tradeoff by consistently showing the lowest or second lowest computation time among other SDP-based methods. Indeed, the aggregation of classes allows us to add more RLT cuts while remaining tractable. Note that the

| | SDP$_T$ | | | | SDP$_U$ (ours) | | | |
|---|---|---|---|---|---|---|---|---|
| **Pruning** | Inactives | | Full | | Inactives | | Full | |
| **Network** | Cert. | Time | Cert. | Time | Cert. | Time | Cert. | Time |
| 6x100 | 74 | 399 | 72 | 131 | 94 | 384 | 86 | 323 |
| 6x200 | 74 | 2109 | 64 | 1367 | 90 | 544 | 85 | 441 |

Figure 2: Impact of the ablation of neurons (stable inactive only or stable inactive and active)

pruning of stable active neurons also has a significant impact on the size of our model ($SDP_U$) (see the proportion of stable active neurons in Table 3), which further accelerates the total computation time.

### 5.3 STABLE ACTIVE NEURONS PRUNING

We now study the impact of the ablation of stable active neurons on the performance of methods SDP$_T$ and SDP$_U$. We ran our experiments on networks 6x100 and 6x200, and we report the results in Table 2. For each method, we consider two cases: pruning of inactive neurons only using 50% of RLT cuts, and pruning of both stable active and inactive neurons. The results reveal a similar trend for both methods. As expected, the greater the number of pruned neurons, the faster the resolution. Performing this ablation along with chordal decomposition allows for drastically reducing the number of variables. The pruning slightly reduces the quality of some constraints, but the impact on the percentage of certification remains limited.

### 5.4 MULTICLASS SCALING

Finally, we assess the scalability of our method with respect to large-scale, multi-class datasets. We constructed a composite dataset by merging EMNIST Balanced, KMNIST, and FashionMNIST, resulting in a total of 67 distinct classes. We trained neural networks on subsets of this dataset, with 5, 20, 50, and 67 classes respectively, with one representative for each class and $\epsilon = 0.05$. We compare the runtime performance of SDP$_U$ against SDP$_T$ and $\beta$-CROWN. We report the results in Figure 6, where each line plots the computation time (in seconds) according to the number of classes and also specifies the certification percentage. We observe that the computation time of SDP$_T$ increases greatly with the number of classes, while the computation time of SDP$_U$ remains stable. Clearly, the aggregation of classes enables a significant speed-up towards large multi-class datasets. Note moreover that while pruning of trivially certified targets with $\beta$-CROWN is useful for scaling, its effectiveness decreases for large number of classes. Furthermore, our approach is the only

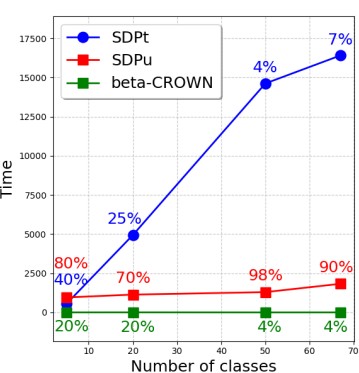

Figure 3: Impact of the number of classes over the certification time and accuracy

one showing satisfying certification scores when the number of classes increases, while the performances significantly drop for ($SDP_T$) and $\beta$-CROWN. This reflects how obtaining good robustness becomes challenging for large multi-class datasets, thus highlighting the relevance of a unified approach accross all classes.

## 6 CONCLUSION

We have introduced a new SDP model to verify ReLU networks across all targets, enabling a significant speedup compared to current SDP models. We are further able to improve both targeted and untargeted models thanks to a reduction in the size of the SDP models by ablation of variables corresponding to stable active neurons. Further work could include the combination of multiclass certification, together with a branch and bound strategy to perform complete verification.

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
