APPENDIX

# A PROOFS

## A.1 PROOF OF PROPERTY 3

*Proof.* Constraint (6) taking $k = 0$ is equivalent to Constraint (3), ensuring that $z_0 \in \mathcal{B}_\epsilon(x)$. Thus, $v(QP_T^j) \geq 0$ implies $\min_{z_0 \in \mathcal{B}_\epsilon(x)} z_K^y - z_K^j \geq 0$. $\qquad\square$

*Proof.*      1. We first prove that $v(QP_T^{\bar{j}}) \geq v(QP_u)$ for all $\bar{j} \in \bar{\mathcal{J}}_K$. Let $z^*$ be an optimal solution of $(QP_t^{\bar{j}})$. We build the solution $(z = z^*, \beta)$ of $(QP_U)$ with $\beta_{\bar{j}} = 1$ and $\beta_j = 0 \quad \forall j \in \bar{\mathcal{J}}_K \setminus \{\bar{j}\}$ satisfying Constraints (19) and (18). Obviously, the two objective functions have the same value.

     2. Then, we prove that $\min_{\bar{j} \in \bar{\mathcal{J}}_K} v(QP_T^{\bar{j}}) \leq v(QP_U)$.

         Let $(z^*, \beta^*)$ an optimal solution of $(QP_U)$ with $\beta_{\bar{j}} = 1$ for $\bar{j} \in \bar{\mathcal{J}}_K$. Obviously, $z = z^*$ is feasible for $(QP_T^{\bar{j}})$. Once again, the objective values are the same.

$\qquad\square$

## A.2 PROOF OF PROPOSITION 1

*Proof.* We ensure that the inequalities are valid in all three possible cases:

     1. If $\beta_{j_1} = 1$ and $\beta_{j_2} = 0$, then both sides of (26) are equal to 0 and (27) becomes $U^{j_1} \geq z_K^{j_1}(x)$.

     2. $\beta_{j_1} = 0$ and $\beta_{j_2} = 1$, (26) and (27) respectively lead to $z_K^{j_2}(x) \leq U_K^{j_2}$ and $z_K^{j_2}(x) \geq z_K^{j_1}(x)$. The latter is valid since when $\beta_{j_2} = 1$, class $j_2$ provides the worst example.

     3. Otherwise, $\beta_{j_1} = \beta_{j_2} = 0$ and we obtain $L_K^{j_1} \leq z_K^{j_1}(x)$ from (26) and $U_K^{j_1} \geq z_K^{j_1}(x)$ from (27).

$\qquad\square$

## A.3 CONSTRAINTS EQ. (13) OR EQ. (9) ALSO IMPLIES $L_k \leq P[z_k] \leq U_k$ IN THE SDP CASE

*Proof.* We want to prove that $P[z_k z_k^T] - (U_k + L_k)P[z_k] + U_k L_k \leq 0$ implies $L_k \leq z_k \leq U_k$. Assume $P$ is positive semi-definite, so every principal minor (ie. every determinant of any principal submatrix) is nonnegative. In particular : $\det \begin{pmatrix} P[1] & P[z_k] \\ P[z_k] & P[z_k z_k^T] \end{pmatrix} \geq 0$, and since $P[1] = 1$, this gives $P[z_k]^2 \leq P[z_k z_k^T]$. Plugging $P[z_k]^2$ into Eq. (13) we obtain $P[z_k]^2 - (U_k + L_k)P[z_k] + U_k L_k \leq 0$, which is equivalent to $(U_k - P[z_k])(P[z_k] - L_k) \geq 0$. $L_k \leq U_k$ gives : $L_k \leq P[z_k] \leq U_k$.

The same reasoning applies with $P_k$. $\qquad\square$

## A.4 PROOF OF PROPOSITION 2

*Proof.* We consider here the case where the neural network has more than one hidden layer, *i.e.*, $K > 2$:

- If $k \in [K-3]$ for $SDP_U$, or if $k \in [K-2]$ for $SDP_T$:
  The matrix $P_k$ in $SDP_T$ and $SDP_U$ is of size $1 + n_k^u + n_k^a + n_{k+1}^u + n_{k+1}^a$, where $n_k^a$ is the number of stable active neurons in layer $k$, and $n_k^u$ the number of unstable neurons. Note that $n_k^a + n_k^u$ corresponds to the layer size $n_k$ minus the number of stable inactive neurons. Pruning the stable active neurons in layer $k$ removes $n_k^a$ rows from the matrix $P_k$. The same applies to layer $k+1$. The final matrix $P_k$ is therefore of size $1 + n_k^u + n_{k+1}^u$. The number of matrix entries thus decreases from $\left(1 + n_k^u + n_k^a + n_{k+1}^u + n_{k+1}^a\right)^2$ to $\left(1 + n_k^u + n_{k+1}^u\right)^2$.

- If $k = K-2$ for $SDP_U$: the matrix size without pruning of stable active neurons is $1 + n_{K-2}^u + n_{K-2}^a + n_{K-1}^u + n_{K-1}^a + |\bar{\mathcal{J}}_K|$. Removing the stable active neurons reduces the matrix size to $1 + n_{K-2}^u + n_{K-1}^u + |\bar{\mathcal{J}}_K|$. The number of matrix entries thus goes from $(1 + n_{K-2}^u + n_{K-2}^a + n_{K-1}^u + n_{K-1}^a + |\bar{\mathcal{J}}_K|)^2$ to $(1 + n_{K-2}^u + n_{K-1}^u + |\bar{\mathcal{J}}_K|)^2$.

If the neural network has one or zero hidden layer, there is only one matrix variable in the model, namely $P_0$. The same reasoning as above can be applied to compute the reduction in the number of matrix entries.

$\square$

## B COMPLEMENTS ON METHOD

### B.1 ALGORITHMS COMPARAISON

---

$\text{SDP}_T\ (\epsilon, x)$
cert ← true;
**for** $j \in \bar{\mathcal{J}}_K$ **do**
    Compute bounds with $\beta$-CROWN.
    **if** $\exists l \in \bar{\mathcal{J}}_K \cup y,\ \ U_K^j < L_K^l$ **then**
        **break**;
    $v_j^* \leftarrow$ Solve $(SDP_T^{\epsilon, j, x})$;
    **if** $v_j^* \leq 0$ **then**
        cert ← false;
        **break**;
**return** cert;

---

**Algorithm 1:** Full certification of $\epsilon$-robustness of a DNN with targeted SDP model $(SDP_T)$ on data $x \in \mathcal{D}$

---

$\text{SDP}_U\ (\epsilon, x)$
cert ← true;
Compute bounds with $\beta$-CROWN.
$T \leftarrow \{j \in \bar{\mathcal{J}}_K, \ \ \exists l \in \bar{\mathcal{J}}_K \cup y, \ \ U_K^j < L_K^l\}$
$v^* \leftarrow$ Solve $(SDP_U^{\epsilon, T, x})$;
**if** $v^* \leq 0$ **then**
    cert ← false;
    **break**;
**return** cert;

---

**Algorithm 2:** Full Certification of $\epsilon$-robustness of a DNN with untargeted SDP model $(SDP_U)$ on data $x \in \mathcal{D}$

### B.2 STABLE ACTIVE NEURONS ABLATION

In this section, we explain with more details our $ReLU$ constraint relaxation in the context of stable active neurons ablation.

For a given $k \in [K-2]$, the quadratic $ReLU$ constraint on neuron $j$ of layer $k+1$ is

$$z_{k+1}^j(z_{k+1}^j - W_{k+1,u}^j z_k^u - b_k) = z_{k+1}^j W_{k+1,a}^j z_k^a$$

where $z_k^a$ is the vector of stable active neurons of layer $k$ and $z_k^u$ is the vector of unstable neurons on layer $k$.

Note that each stable active neuron $r$ of layer $k$ can be decomposed into a linear combinations of previous layers outputs $z_k^r = \sum_{l=0}^{k-1} \sum_{i=1, i \text{ unstable}}^{n_l} \lambda_l^i z_l^i + \gamma$, where $\gamma \in \mathbb{R}$, $\lambda_l^i \in \mathbb{R} \ \forall l \in [0, k-1]$, $i \in [1, n_l]$. This decomposition can be computed dynamically as follows :

- for a layer $k = 1$ and a stable active neuron $r$ on such layer: $z_1^r = W_1^r z_0 + b_1^r$;

- for a layer $k > 1$ and a stable active neuron $r$ on such layer,

$$
\begin{aligned}
z_k^r &= W_k^r z_{k-1} + b_k^r \\
&= W_{k,u}^r z_{k-1}^u + W_{k,a}^r z_{k-1}^a + b_k^r \\
&= W_{k,u}^r z_{k-1}^u + \sum_{\nu=1,\nu\text{ active}}^{n_{k-1}} W_{k,\nu}^r \Big( \sum_{l=0}^{k-2} \sum_{i=1,i\text{ unstable}}^{n_l} \lambda_{l,\nu}^i z_l^i + \gamma_\nu \Big) + b_k^r \\
&= W_{k,u}^r z_{k-1}^u + \sum_{l=0}^{k-2} \sum_{i=1,i\text{ unstable}}^{n_l} \Big( \sum_{\nu=1,\nu\text{ active}}^{n_{k-1}} W_{k,\nu}^r \lambda_{l,\nu}^i \Big) z_l^i + \sum_{\nu=1,\nu\text{ active}}^{n_{k-1}} W_{k,\nu}^r \gamma_\nu + b_k^r
\end{aligned}
$$

Finally, our *ReLU* quadratic constraint can be rewritten

$$
z_{k+1}^j (z_{k+1}^j - W_{k,u}^j z_k^u - b_k) = \sum_{l=0}^{k-1} \sum_{i=1,i\text{ unstable}}^{n_l} A_l^i z_{k+1}^j z_l^i + B z_{k+1}^j \tag{32}
$$

where $A$ is derived from the product of $W_{k+1}^a$ and the coefficients $\lambda$, and B is derived from the product of $W_{k+1}^a$ and the coefficients $\gamma$.

We now consider the quadratic term $z_{k+1}^j z_l^i$ present in the non-relaxed *ReLU* constraint on neuron $j$ of layer $k+1$, where $l$ is a previous layer $l \in [k-1]$ and $i$ is a neuron on such layer. Due to the chordal decomposition, only products of neurons of two consecutive layers are considered and the quadratic terms $z_{k+1}^j z_l^i$ do not appear in our matrix variables. Our idea is to bound the terms $z_{k+1}^j z_l^i$ using the upper and ower bounds given by McCormick envelopes (McCormick, 1976). We have the following bounding inequalities $0 \le z_{k+1}^j \le U_{k+1}^j$, $\tilde{L}_l^i \le z_l^i \le U_l^i$, where:

- $U_{k+1}^j$ and $U_l^i$ are computed with $\beta$-CROWN.
- $\tilde{L}_l^i$ are computed as follows:
    - For the input $l = 0$, we take the known trivial lower bounds : $\tilde{L}_l^i = L_0^i = x^i - \epsilon$;
    - For hidden layers $l > 0$, as all stable neurons have been pruned, neuron $i$ of layer $l$ is unstable, that is to say its lower bound computed by $\beta$-CROWN is negative : $L_l^i < 0$. As $z_l^i$ represents the post-activation value and we want to take the tightest lower bound possible to obtain the most efficient cuts, we take : $\tilde{L}_l^i = 0$.

This enables us to formulate the McCormick envelopes:

$$
\begin{cases}
z_{k+1}^j z_l^i \ge \tilde{L}_l^i z_{k+1}^j \\
z_{k+1}^j z_l^i \le U_l^i z_{k+1}^j \\
z_{k+1}^j z_l^i \ge U_l^i z_{k+1}^j + U_{k+1}^j z_l^i - U_{k+1}^j U_l^i \\
z_{k+1}^j z_l^i \le \tilde{L}_l^i z_{k+1}^j + U_{k+1}^j z_l^i - U_{k+1}^j \tilde{L}_l^i
\end{cases}
$$

These inequalities yield two boundings of $A_l^i \, z_{k+1}^j \, z_l^i$, the first one being :

$$
A_l^i \, z_{k+1}^j \, z_l^i \in
\begin{cases}
[A_l^i \, \tilde{L}_l^i \, z_{k+1}^j, \; A_l^i \, U_l^i \, z_{k+1}^j] & \text{if } A_l^i \ge 0 \\
[A_l^i U_l^i z_{k+1}^j, \; A_l^i \tilde{L}_l^i z_{k+1}^j] & \text{otherwise.}
\end{cases}
\tag{33}
$$

And the second one being :

$$
A_l^i z_{k+1}^j z_l^i \in
\begin{cases}
[A_l^i\big(\tilde{L}_l^i z_{k+1}^j + U_{k+1}^j z_l^i - U_{k+1}^j \tilde{L}_l^i\big), \; A_l^i\big(U_l^i z_{k+1}^j + U_{k+1}^j z_l^i - U_{k+1}^j U_l^i\big)] & \text{if } A_l^i \ge 0 \\
[A_l^i\big(U_l^i z_{k+1}^j + U_{k+1}^j z_l^i - U_{k+1}^j U_l^i\big), \; A_l^i\big(\tilde{L}_l^i z_{k+1}^j + U_{k+1}^j z_l^i - U_{k+1}^j \tilde{L}_l^i\big)] & \text{otherwise.}
\end{cases}
\tag{34}
$$

By summing up the lower linear bounds of $A_l^i z_{k+1}^j z_l^i$ in (33) for all $l \in [0, k-1], i \in [1, n_l]$, we obtain a lower linear bound on the right term of (32) : $\sum_{l=0}^{k-1} \sum_{i=1}^{n_l} A_l^i z_{k+1}^j z_l^i + B z_{k+1}^j \leq (C_1 + B) z_{k+1}^j$.

Applying the same approach for the upper bound of (33) and lower and upper bound of (34), we obtain our final set of 4 constraints:

$$
\begin{cases}
z_{k+1}^j(z_{k+1}^j - W_{k,u}^j z_k^u - b_k) \leq (C_1 + B) z_{k+1}^j & 28 \\[2mm]
z_{k+1}^j(z_{k+1}^j - W_{k,u}^j z_k^u - b_k) \geq (C_2 + B) z_{k+1}^j & 29 \\[2mm]
z_{k+1}^j(z_{k+1}^j - W_{k,u}^j z_k^u - b_k) \leq \sum_{l=0}^{k-1} \sum_{i=1}^{n_l} C_{3,l}^i z_l^i + C_{3,k+1} z_{k+1}^j + C_3 & 30 \\[2mm]
z_{k+1}^j(z_{k+1}^j - W_{k,u}^j z_k^u - b_k) \geq \sum_{l=0}^{k-1} \sum_{i=1}^{n_l} C_{4,l}^i z_l^i + C_{4,k+1} z_{k+1}^j + C_4 & 31
\end{cases}
$$

where

$$
\begin{cases}
C_1 = \sum_{l=0}^{k-1} \sum_{i=1, A_l^i \geq 0}^{n_l} A_l^i U_l^i + \sum_{l=0}^{k-1} \sum_{i=1, A_l^i < 0}^{n_l} A_l^i \tilde{L}_l^i \\[3mm]
C_2 = \sum_{l=0}^{k-1} \sum_{i=1, A_l^i < 0}^{n_l} A_l^i U_l^i + \sum_{l=0}^{k-1} \sum_{i=1, A_l^i \geq 0}^{n_l} A_l^i \tilde{L}_l^i \\[3mm]
C_3 = B - \sum_{l=0}^{k-1} \sum_{i=1, A_l^i \geq 0}^{n_l} A_l^i U_{k+1}^j U_l^i - \sum_{l=0}^{k-1} \sum_{i=1, A_l^i < 0}^{n_l} A_l^i \tilde{L}_l^i U_{k+1}^j \\[3mm]
C_4 = B - \sum_{l=0}^{k-1} \sum_{i=1, A_l^i \geq 0}^{n_l} A_l^i U_{k+1}^j \tilde{L}_l^i - \sum_{l=0}^{k-1} \sum_{i=1, A_l^i < 0}^{n_l} A_l^i U_l^i U_{k+1}^j \\[3mm]
C_{3,k+1} = \sum_{l=0}^{k-1} \sum_{i=1, A_l^i \geq 0}^{n_l} A_l^i U_l^i + \sum_{l=0}^{k-1} \sum_{i=1, A_l^i < 0}^{n_l} A_l^i \tilde{L}_l^i \\[3mm]
C_{4,k+1} = \sum_{l=0}^{k-1} \sum_{i=1, A_l^i \geq 0}^{n_l} A_l^i \tilde{L}_l^i + \sum_{l=0}^{k-1} \sum_{i=1, A_l^i < 0}^{n_l} A_l^i U_l^i \\[3mm]
C_{3,l}^i = C_{4,l}^i = A_l^i U_{k+1}^j & \forall i \in [1, n_l], l \in [0, k-1]
\end{cases}
$$

Note that a wider set of constraints could be explored by varying combinations of bounds from (33) and (34). This ablation multiply by 4 the number of *ReLU* constraints dedicated to unstable neurons. However, it eliminates several categories of constraints related to stable active neurons, including the *ReLU* constraint (11), the bounding constraints (13), the triangular constraints (14), the McCormick constraints (24), and the RLT constraints (16). Notably, this leads to a quadratic reduction with respect to $n_k^a$: $4\, n_k^a\, |\bar{\mathcal{J}}_K|$ constraints removed from 24, and up to $n_k^a(n_k^u + n_k^a)$ constraints removed from (16) on layer $k$. In sufficiently big neural networks, this quadratic reduction counterbalances the linear increase in constraints with respect to $n_k^u$.

## B.3 TRIANGULAR CONSTRAINT

We say that a neuron $j$ of layer $k$ of lower bound $L_k^j$ and upper bound $U_k^j$ is *stable active* if $L_k^j \geq 0$ and *stable inactive* if $U_k^j \leq 0$. Otherwise, a neuron is *unstable* (i.e. if $L_k^j < 0 < U_k^j$).

In order to tighten the upper bound on the *ReLU* activation function, a well-known constraint is the triangular constraint, which provides a convex embedding of the *ReLU* output. Depending on the

activation status of the considered neuron $j$ of layer $k$, we decompose the linear upper bound in the following set $\mathcal{T}$ of triangular inequalities:

$$z_k^j \in \mathcal{T} \Leftrightarrow \begin{cases} z_k^j \leq 0 & \text{if } j \text{ is } \textit{inactive} \\ z_k^j \leq W_k^j z_{k-1} + b_k^j & \text{if } j \text{ is } \textit{active} \\ z_k^j \leq \dfrac{U_k^j}{U_k^j - L_k^j}(W_k^j z_{k-1} + b_k^j) \\ \qquad + \dfrac{U_k^j}{U_k^j - L_k^j}(b_k^j - L_k^j) & \text{if } j \text{ is } \textit{unstable} \end{cases}$$

Combined with constraints $z_k^j \geq 0$, $z_k^j \leq \hat{z}_k^j$, where $\hat{z}_k^j$ denotes preactivation vector of neuron $j$ of layer $k$, this constraint yields the exact output $z_k^j = 0$ when neuron $j$ is inactive, and $z_k^j = W_k^j z_{k-1} + b_k^j$ when it is active.

When neuron $j$ is unstable, the upper-bound is plotted in red in Figure 4. The equation of the red line

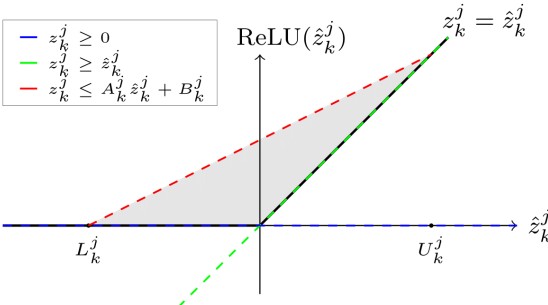

Figure 4: Triangular constraint on neuron $j$ of layer $k$, where $\hat{z}_k^j = W_k^j z_{k-1} + b_k^j$ is the pre-activation vector

clearly depends on the lower and upper bounds $L_k^j$ and $U_k^j$ of the preactivation vector $W_k^j z_{k-1} + b_k^j$.

The triangular constraint has been shown to be limited, as it represents only the convex hull of the output of a single *ReLU* neuron (Salman et al., 2019). Recent works have porposed convex relaxations to capture the joint behavior of multiple *ReLUs*. Such ideas could be explored to compute more efficient cuts in SDP models.

### B.4 COHERENCE CONSTRAINT EQ. (15)

In all our SDP models, we use the constraint between two consecutive matrices $P_k[(1 \ z_{k+1})(1 \ z_{k+1})^\top] = P_{k+1}[(1 \ z_{k+1})(1 \ z_{k+1})^\top]$ (15) relaxed as in (Batten et al., 2021) and in (Lan et al., 2022). For a layer $k$, including all coherence constraints in the model would introduce $\frac{n_k(n_k+3)}{2}$ constraints. In the relaxation of constraint (15), only the $n_k$ linear constraints remain: $P_k[z_{k+1}] = P_{k+1}[z_{k+1}]$, preventing the number of constraints from exploding.

## B.5 RLT CONSTRAINT

We present here the RLT cuts (16) selected in (Lan et al., 2022). These cuts contribute to tightening the relaxation, and are given below:

$$\text{Tightening cuts}\begin{cases} z_k z_{k+1} \leq L_k z_{k+1} - U_{k+1} z_k + L_k U_{k+1} & (35) \\ z_k z_{k+1} \leq L_k z_{k+1} - U_{k+1} z_k + L_k U_{k+1} & (36) \\ z_k z_{k+1} \leq L_k z_{k+1} - L_{k+1} z_k + L_k L_{k+1} & (37) \\ P_k[z_{k+1} z_{k+1}] \leq U_{k+1} z_{k+1} & (38) \\ P_k[z_{k+1} z_{k+1}^T] \leq A_k P_k[z_k z_{k+1}] + B_k P_k[z_{k+1}] & (39) \end{cases}$$

The number of these constraints is large. For a given layer $k$, constraints (35)–(37) scale quadratically with $n_{k+1} \times n_k$, while constraints (38) and (39) scale with $n_{k+1} \times n_{k+1}$. Including all these constraints in the SDP model would significantly increase the computing time. A heuristic is therefore needed to select only a subset of these cuts. As (38) and (39) capture *intra* layer dependencies, a heuristic selecting a subset of them is difficult to design. In contrast, since (35)–(37) represents *inter* layer dependencies, a heuristic based on the linear layer weights linking them is possible.

Only a subset of constraints (35)–(37) is finally selected, based on a given percentage $p$. Specifically, for each neuron $j$ on layer $k + 1$ we select $\lfloor p\, n_k \rfloor$ cuts. The heuristic sorts the absolute value of the weights $|W_{k+1}^j|$, and selects the neurons corresponding to the top $\lfloor p\, n_k \rfloor$ entries in the sorted vector. This selection is based on the full size of layer $k$ regardless of whether neurons have been pruned. More precisely, in the context of a full ablation, we do not select $\lfloor p\, n_k^u \rfloor$ RLT cuts but $\lfloor p\, n_k \rfloor$ ones.

## B.6 TIGHTENING CUTS FOR SDP$_U$

For simplicity and clarity, we used the full logits $z_K^j$ in constraints (26) (27) (24). Note that these logits are not variables of our model. To obtain the full constraints in our model, we need to substitute each logit by its linear expression with respect to the penultimate layer variables : $z_K^j = W_K^j z_{K-1} + b_K^j$.

Furthermore, for the sake of clarity in constraints (23) (24) (26) (27 (25), we omit explicit matrix indexation. To recover the full constraint, one must, for example, replace variables such as $\beta_i \beta_j$ by $P_{K-2}[\beta_i \beta_j]$. Combining this with the logit expression, $z_K^j$ should be replaced by $W_K^j P_{K-2}[z_{K-1}] + b_K^j$, and $\beta_j z_K^j$ by $W_K^j P_{K-2}[\beta_j z_{K-1}] + b_K^j P_{K-2}[\beta_j]$.

## C    EXPERIMENTATIONS

### C.1    STUDY OF THE IMPACT OF RLT CUTS

We study the influence of the percentage of RLT cuts considered (parameter $p$ of the heuristic of (Lan et al., 2022)) on the CPU times and the certification accuracy. For a small percentage of RLT cuts, the targeted ($SDP_T$) model certifies better, while for $p \geq 70\%$ untargeted ($SDP_U$) gives a better bound. Moreover, the computing time of ($SDP_U$) increases significantly slower than that of ($SDP_T$) with the addition of RLT cuts.

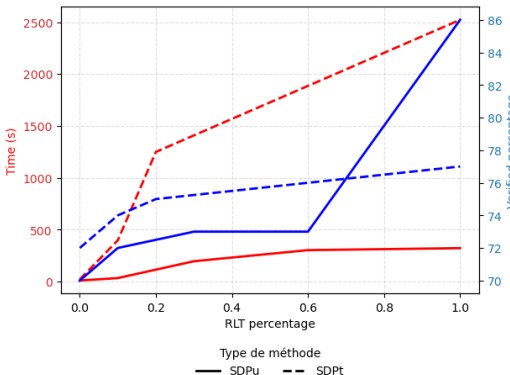

Figure 5: Impact of the RLT percentage on models performance on the 6x100 network. Certification percentage in blue and computing time (s) in red.

### C.2    STUDY OF THE PERTURBATION SIZE

We evaluate the impact of the attack ball radius $\epsilon$ on our SDP formulation. Running the models on the 6x100 neural network with $\epsilon \in \{0.026, 0.05, 0.1\}$, we see that ($SDP_U$) gives even better certification results as the perturbation size increases. The efficiency of our method is both due to the tightness of SDP bounds and to the large number of RLT cuts that our model is able to handle ($p = 100\%$). Note that for increasing $\epsilon$, the size of ($SDP_T$) increases even more rapidly than that of ($SDP_U$) as it does not prune stable active neurons. The under-performing behavior of $\beta$-CROWN may be due to the over-conservativeness of its bounds.

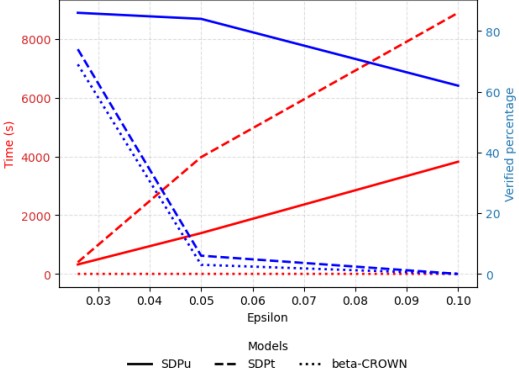

Figure 6: Impact of the perturbation size on models performance on the 6x100 network. Certification percentage in blue and computing time (s) in red.

### C.3    IMPLEMENTATION DETAILS

All networks have been trained with a batch size of 128, the Adam optimizer, and a learning rate of 0.001. For reproducibility, we show the details of the adversarial training in table 2. We used

the optimizer Adam, a learning rate $lr = 0.01$, and batch sizes of 128. All PGD attacks were used with number of steps $= 40$, a random start. We denote by `6x100-5`, `6x100-20`, `6x100-50`, `6x100-67` networks used in experiment 3.

| Network | Architecture | Adversarial Training | | Accuracy |
|---|---|---|---|---|
| | | Epochs | Adversarial attack | |
| 6x100 | 784-6x100-10 | 200 | PGD ($\epsilon = 0.3$, $\alpha = 0.01$) | 95.5 |
| 6x200 | 784-6x200-10 | 200 | PGD ($\epsilon = 0.3$, $\alpha = 0.01$) | 96.1 |
| 9x100 | 784-9x100-10 | 200 | PGD ($\epsilon = 0.3$, $\alpha = 0.01$) | 95.2 |
| 9x200 | 784-9x200-10 | 200 | PGD ($\epsilon = 0.3$, $\alpha = 0.01$) | 96.9 |
| 6x100-5 | 784-6x100-5 | 100 | PGD ($\epsilon = 0.3$, $\alpha = 0.01$) | 97.6 |
| 6x100-20 | 784-6x100-20 | 100 | PGD ($\epsilon = 0.3$, $\alpha = 0.01$) | 86.0 |
| 6x100-50 | 784-6x100-50 | 100 | PGD ($\epsilon = 0.3$, $\alpha = 0.01$) | 76.7 |
| 6x100-67 | 784-6x100-67 | 100 | PGD ($\epsilon = 0.3$, $\alpha = 0.01$) | 75.2 |

Table 2: Networks used in our three experiments.

| Network | $\epsilon$ | Stability study | | | | Running targets |
|---|---|---|---|---|---|---|
| | | Stable | | Unstable | Total | |
| | | Inactives | Actives | | | |
| 6x100 | 0.026 | **283** | **224** | **93** | **600** | 4.0 |
| | | 47% | 37% | 16% | 100% | |
| 6x200 | 0.015 | **713** | **287** | **200** | **1200** | 6.8 |
| | | 59% | 24% | 17% | 100% | |
| 9x100 | 0.026 | **364** | **225** | **312** | **900** | 7.1 |
| | | 40% | 25% | 35% | 100% | |
| 9x200 | 0.015 | **1024** | **312** | **464** | **1800** | 7.6 |
| | | 57% | 17% | 26% | 100% | |
| 6x100-5 | 0.05 | **261** | **119** | **220** | **600** | 3.25 |
| | | 44% | 20% | 36% | 100% | |
| 6x100-20 | 0.05 | **243** | **157** | **201** | **600** | 14.44 |
| | | 40% | 26% | 33% | 100% | |
| 6x100-50 | 0.05 | **188** | **117** | **295** | **600** | 48.8 |
| | | 31% | 20% | 49% | 100% | |
| 6x100-67 | 0.05 | **203** | **127** | **271** | **600** | 60.1 |
| | | 34% | 21% | 45% | 100% | |

Table 3: Mean number of stable active, stable inactive and unstable neurons computed on the data of our experiments (100 data for `6x100-6x200-9x100-9x200`, 5 data for `6x100-5`, 20 data for `6x100-20`, 50 data for `6x100-50`, and 67 data for `6x100-67`).

## D ADDITIONNAL STATE OF THE ART

### D.1 STANDARD SDP RELAXATIONS

The two main criterion to design a relaxation is its computation time and its tightness. Indeed, the tighter the relaxation, the better the lower bound. Existing relaxation techniques for quadratic problems are mainly based on linearization or on semi-definite programming. To compute a linear relaxation, the quadratic functions are reformulated as convex equivalent functions in an extended space of variables. More precisely, new variables $Z_{ij}$ are introduced for all $(i, j)$, that are meant to satisfy the equalities $Z_{ij} = z_i z_j$. The linearization is then obtained by relaxation of the later non-convex equalities, for instance by linear constraints (see for instance (McCormick, 1976; Sherali & Adams, 2013; Yajima & Fujie, 1998)). Using semi-definite relaxations for quadratic programming was also widely studied (Anstreicher, 2009; Chen & Burer, 2012; Burer & Vandenbussche, 2008; 2009; Vandenbussche & Nemhauser, 2005a;b). A semi-definite relaxation of a quadratic optimization problem can be obtained by lifting $z$ to a symmetric matrix $Z = zz^\top$ where the later non-convex constraints are relaxed to $Z - zz^\top \succeq 0$. Note that, since in a DNN only layers $k$ and $k+1$ are linked by $ReLU$ Constraints, a chordal decomposition of the variable matrix $Z$ into $K-1$ block diagonal matrices is possible (Batten et al., 2021). This standard semi-definite relaxation is often referred to as "Shor's" relaxation. In (Anstreicher, 2009), the "Shor's plus RLT" relaxation was introduced, where the convex envelopes of the quadratic terms (McCormick, 1976) where added to the later relaxation. We detailed the RLT cuts used in (Lan et al., 2022) in section B.5. Other works have generalised the heuristic for the selection of RLT cuts to create an iterative algorithms, adding iteratively more strategic cuts to strengthen the bound (Lan et al., b). Some works have explored the stability of SDP models for verification (Ueda et al.).

### D.2 COMPLETE VERIFICATION

Ideal verification is complete, ensuring that all answers are reliable. However, due to the complexity of the problem, fully achieving such verification is often constrained. Works using Satisfiability Modulo Theory (Ehlers, 2017) like ReLUplex (Katz et al., 2017) or Marabou (Katz et al., 2019) (Wu et al., 2024) have been developed, but are not currently scalable. Nevertheless, they are very precise and give formal proof of robustness or useful counterexamples when working on a sufficiently small network.

Some works have introduced Mixed Integer Programming formulations (MIP), see (Fischetti & Jo, 2018) (Cheng et al., 2017), but the direct resolution of these models without relaxations are also not scalable. Some works propose an efficient selection of the variables that should be relaxed (Liao et al.). Most efficient complete verification rely on Branch&Bound (Bunel et al., 2020)- (Ferrari et al., 2022)- (Jaeckle et al., 2021)- (Lu & Kumar, 2019), whose relaxation of the certification problem is fast heuristics like CROWN. They have been improved by smart branching: splitting on the activation or not activation of a set of neurons has been exponentially faster than splitting on the input ball. Some methods have improved $ReLU$ splitting to better choose neurons for branching decision (Henriksen & Lomuscio, 2021; Zhou et al., 2024). When reaching a certain depth of the tree, a relatively fast MIP is solved (with few binary variables as the activation of most neurons is fixed), to prune a branch without exploring all its content. It also helps to avoid impossible activation patterns in practice, which may not be seen by heuristic methods like bound propagation, guaranteeing the soundness of the algorithm. The resolution of these MILP has been further improved with cutting planes in method GCP-CROWN (Zhang et al., 2022). Other works are Branch&Bound frameworks based on SDP relaxations (Lan et al., a; Chiu et al., 2025). Note that these approaches primarily improve certification rates rather than scalability.

### D.3 ADVERSARIAL TRAINING

In this section, we present the adversarial training used in order to create robust networks. Madry introduced adversarial training (Madry et al., 2019) by adding a maximisation problem into the common training minimisation problem:

$$\min_{\forall (x,y) \in \mathcal{X}} \max_{z \in \mathcal{B}_\varepsilon(x)} \mathcal{L}(z, y)$$

where the inner maximisation represents the computation of the worst adversarial attack. It is approached by heuristics, most of them are based on gradient descents, Projected Gradient Descent (PGD) (Madry et al., 2019), or its variants (FGSM (Goodfellow et al., 2015), IGS (Kurakin et al., 2017)), JSMA (Papernot et al., 2015) for the norm $\|\|\|_0$, DeepFool (Moosavi-Dezfooli et al., 2016) for $\|\|\|_2$) and have been improved since (Carlini & Wagner, 2017). Specifically, we train our model using the most classic adversarial attack: PGD, and compute untargeted adversarial attacks with it.