# OpenReview forum: "Fast SDP certification of neural networks : towards large multi-class datasets"
_ICLR.cc/2026/Conference — Submitted to ICLR 2026_

### Official Review · Reviewer_4Po3 · 2025-10-19

**Soundness:** 2
**Presentation:** 2
**Contribution:** 2
**Rating:** 4
**Confidence:** 4

**Summary:**

The paper proposes a single (untargeted) quadratic certification model by introducing selector binaries $\beta_j$ so that one optimization certifies robustness across all possible targets, and then derives an SDP relaxation to solve it.  To improve scaling, the authors use chordal decompositions, pruning of stable active (and inactive) neurons and compensating for the missing quadratic couplings via McCormick-based cuts. Experiments on MNIST fully connected feed-forward networks and a 67-class composite benchmark suggest higher certified rates and/or faster runtime with respect to other target-based SDP baselines.

**Strengths:**

Considered problem is relevant, as robust DNN-driven systems are crucial for successful deployment. The pruning of *active* neurons (not only inactive ones) and the chordal decomposition are sensible choices to reduce PSD block sizes in practice.

**Weaknesses:**

Modeling disjunction via binary variables is a relatively standard and well-known approach. The contribution lies more in the SDP instantiation and pruning/cut design, but exhaustive ablation studies illustrating the clear impact of adding specific individual features to the SDP relaxation are missing.

### Major points

- Experimental results are very limited. Only relatively small feed-forward networks are used (up to 1800 hidden neurons). Tightness of the bound and/or scalability of the method with respect to different sizes of perturbation regions is not assessed.
- Presentation: The paper is difficult to follow because key ideas are scattered, notation is overloaded and inconsistent (e.g., superscripts can refer to vectors or specific neurons, $u$ can mean both untargeted and unstable, and many others), with many typos and language issues (e.g., equality in lines 336–337 is mathematically unsound). Explanations for understanding experimental results only come after a few paragraphs or subsections, etc. Overall, a thorough rewrite is necessary before the contributions can be clearly and fairly assessed.

### Minor points

- For most of the references, `\citep{}` should be used to facilitate reading.
- Be consistent in writing. For instance, formulations *ReLU*, *relu*, and *Relu* all appear in the manuscript. Same with *semi-definite*, *semidefinite*, and *semi definite*. The dataset is sometimes denoted $\mathcal{D}$ and sometimes $\mathcal{X}$. Introducing $\mathcal{K}$ is probably not necessary.
- Figure 1 is never referred to in the main text.
- Property 3 is trivial and already explained in words before being formally written down. I suggest removing it. The same can be said about Theorem 1.

**Questions:**

1. Why consider a manufactured dataset with 67 classes rather than a real multiclass dataset (e.g., CIFAR-100 with small convolutional networks)?

2. Wouldn't an explicit proof (even in the Appendix) of Proposition help readers understand the pruning method and, consequently, the structure of the relaxation?

3. What do the percentages in Figure 2 represent?

4. Why not formulating the untargeted objective for other approaches as well and comparing them under this common objective? This would isolate the impact of each modeling choice.

5. For targeted approaches from the literature, how is the target index chosen? Are you solving for all possible targets? Please clarify.

6. For your proposed relaxations, is it possible to analyze loss in bound tightness as a function of, say, attack magnitude, network depth, or fraction of pruned neurons? This analysis could strengthen the paper.

Finally, Your approach selects $\lfloor pn_k\rfloor$ inter-layer RLT cuts per neuron using the top magnitudes $|W^{j}_{k+1}|$,
 computed with $n_k$.

Why is this preferable to selecting $\lfloor p n_k^{u}\rfloor$? What is the sensitivity of certification rate and runtime w.r.t. $p$? Is there an empirical threshold $p_{\min}$ below which tightness and performance degrade notably?

---

> ### Author Response · Authors · 2025-11-20
>
> We thank the reviewer for his insightful remarks. We followed the same validation protocol as in [Lan 2022] which is standard for SDP-based certification methods. To further assess our approach, we varied perturbation sizes on neural network 6x100 and obtain the following results:
>
> | Model | ε = 0.026 | ε = 0.05 | ε = 0.01|
> |--------|---------------|----------------|---------------|
> | $SDP_u$  |      86% - 323s         |    84% - 1391s            |    62% - 3820s            |
> | $SDP_t$  |      74% - 399s         |    6% - 3977s            |    0% - 8890s            |
>
>
> Overall our method remains good in verification percentage, even for great epsilons, while current untargeted models do not scale to great perturbation sizes.
> Furthermore, we analyzed the (SDP) model with RLT fixed at 50% with several values of epsilon (see the following table) for a larger network with 8 hidden layers of size 1024. The results reveal the expected trend: the larger the network and the perturbation, the lower is the number of constraints when full pruning is applied.
>
> |  ε | Pruning| Nb constraints | Nb variables |
> |--------|---------------|----------------|---------------|
> | 0.01 |    Partial      |    7 484 566            |    17 364 036            |
> | 0.01  |     Full    |   7 082 923          |    16 805 075         |
> | 0.1  |     Partial   |   12 299 601          |    32 548 142       |
> | 0.1  |     Full    |   12 299 621         |    32 548 069      |
>
>
> We thank the reviewer for his careful reading. We have carefully checked the final version and corrected the final version. As suggested, we also reorganize the presentation of the results to clarify the contribution. We have corrected the consistency of the writing and the double notation for the dataset. We also removed the notation K. We added a reference to Figure 1 in the final version. We agree that Property 3 and Theorem 1 are explained in words before being formally written, however to facilitate understanding we choose to keep Property 3 and Theorem 1 in the final version.

---

> ### Author Response · Authors · 2025-11-20
>
> ### Questions
>
> *(1)* Fully connected neural networks on CIFAR10 have sometimes been used ([Lan, 2023]), however they do not reach a good clean accuracy (less than 40%). Their performances are thus too erratic to be certified.
>
> *(2)* We give here the main ideas : By denoting $n_k^{a}$ the number of stable active neurons on layer k, and $n_k^u$ the number of unstable neurons, when applying the chordal decomposition each block matrix $P_k$ (k=0 .. K-3) is a symmetric matrix of dimension (1+ $n_k^u$  + $n_k^a$ + $n_{k+1}^u$  + $n_{k+1}^a$), by removing the stable neurons theses sizes become (1+ $n_k^u$ + $n_{k+1}^u$ ). Now for the matrix $P_{K-2}$ that is composed of elements z and of beta (with a number |$J_K$|) , the new size goes from  (1+ $n_{K-1}^u$  + $n_{K-1}^a$ + $n_{K}^u$  + $n_{K}^a$ + |$J_K$|) to  (1+ $n^{K-1}_u$ + $n^{K}_u$ + |$J_K$|). We will add this explanation in the appendix to clarify the proposition.
>
> *(3)* The percentages represent the verification percentage obtained on the data run on. We see that our model is able to certify a significant amount of data even when the number of classes of the dataset increases. For the 50 classes and 67 classes points in our graph, we initially ran ($SDP_t$) only on 5 data due to the high resolution time. We ran it thoroughly and finally obtained 4% and 7% of certification respectively.
>
> *(4)* Our contribution is divided into three parts : (i) an exact untargeted quadratic formulation and its SDP relaxation, (ii) tightening cuts, and (iii) a stable active neuron ablation. We would like to highlight that $SDP_u$ corresponds to the untargeted objective applied to the model  ($SDP_t$), for which we add additional cuts.
> We also ran our model $SDP_u$ without any RLT cuts, which is the application of the  untargeted objective to (SDP_Layer) on network 6x100, and we get a certification percentage of 70% with a mean computing time of  11s.
>
> *(5)* Thank you for raising this point. In other approaches from literature, the choice of an adversarial target has to be made which might bias the results. However, in our paper, we run all models to certify across all targets. At the end, for each targeted method, the number of runs is equal to the number of evaluated data times the number of classes.
>
> *(6)* We have conducted experiments to evaluate the percentage of verified data as a function of the number of stable neurons. We observe that when the proportion of active stable neurons ranges between 50% and 70%, the number of certified data points is significantly higher when pruning the active neurons. For larger proportions, pruning degrades the certification. The study of the impact magnitude is shown above (see answer to question 1).
> Finally, experiments in Tab. 1 show a global trend with respect to the network depth. Indeed, as the depth increases (from 6 to 9 hidden layers), certification performances deteriorate greatly for the baselines while ours remain stable and the computing time stays controlled.
>
> *(7)* We use the same heuristic as in [Lan 2022] to select the RLT cuts. Furthermore, we selected $\lfloor p n_k \rfloor$ cuts in our models to have a final number of RLT cuts similar to $SDP_t$ when having the same value of $p$. In our study on the 6x100 neural network there is not threshold below which performances degrade notably but rather a threshold above which performances improve (from 60% RLT cuts to 100% RLT cuts).

---

> > ### Comment · Reviewer_4Po3 · 2025-11-24
> > **First response to the authors' rebuttal**
> >
> > Dear Authors,
> >
> > Thank you for your efforts in clarifying several of the raised points. However, the currently available version of the manuscript appears to be identical to the initial submission.
> >
> > Would it be possible for you to provide the revised and improved version, with the major changes clearly highlighted (in blue)? This would allow me to better assess the quality of your responses and to reevaluate the strength of the submission. I believe the other Reviewers would also find this helpful.
> >
> > Thank you in advance.

---

> ### Author Response · Authors · 2025-11-25
>
> Thank you for your message and for the careful evaluation of our work.
>
> We would like to confirm that we have now uploaded the revised version of the paper (and appendix). All your comments as well as those from the other reviewers have been carefully addressed, and the corresponding modifications are highlighted in blue.
>
> Please do not hesitate to let us know if anything is missing or if further clarification would be helpful.
> We sincerely appreciate your time and constructive feedback.

---

### Official Review · Reviewer_WUED · 2025-10-24

**Soundness:** 2
**Presentation:** 2
**Contribution:** 2
**Rating:** 4
**Confidence:** 4

**Summary:**

This paper aims to accelerate incomplete verification of adversarial robustness by (i) casting an untargeted single quadratic objective that accounts for all classes, (ii) deriving a semidefinite relaxation augmented with tightening cuts, and (iii) introducing a pruning rule for provably stable-active neurons to reduce computational load.

**Strengths:**

(1) The move from per-class targeted certification to a single untargeted quadratic program with class-selection binaries $\beta_j$ is clear. Theorem 1 shows equivalence to minimizing over targets, justifying the single-SDP approach.
(2) Tight, well-motivated relaxation: The SDP includes chordal decomposition, McCormick envelopes for $\beta \times z$, and two new coupling inequalities ((26)–(27)) tailored to class selection; these are technically sound choices for tightening.
(3) The paper prunes stable active ReLU units and replace them with linear expressions, then compensates the missing quadratic terms with linear bounds. This is practical contribution with qualification in Proposition 2.
(4) The results in Figure 2 strongly demonstrate the scalability with respect to the number of classes.

**Weaknesses:**

(1) RLT cuts are formed by multiplying valid linear inequalities to obtain quadratic ones; as the number of RLT constraints grows, the candidate set can blow up combinatorially, risking intractability. The paper chooses a cut percentage  $p$ by heuristic per architecture. Please report certificate rate and runtime as functions of  $p∈(0,10,30,60,100)$% and identify the “sweet-spot” $p^*$.

(2) Only fully connected networks are evaluated. Extensions to CNNs/ResNets (where SDP relaxations are common) are missing. Even a small CNN/ResNet ablation would clarify whether the method’s gains persist under weight sharing and deeper topologies.

(3) Baseline coverage is weak. Beyond SDP, include LP relaxations, $β$-CROWN, and other established incomplete verifiers, under matched perturbation radii. Please sweep multiple $ϵ∈(1/255,2/255,4/255,8/255)$ and report certified accuracy–runtime curves.

(4) Section 5.2 claims that $\beta-$CROWN  is unable to tighten the bounds to eliminate all possible target classes as the number of neurons increases, but no experiment demonstrates this trend. Provide plots of (i) average margin lower bound vs. width/depth, and (ii) fraction of eliminated classes vs. network size.

(5) Table 2 shows fewer certified cases with pruning, but it remains unclear why. Please quantify: (i) certified fraction vs. number of unstable ReLUs, (ii) average bound gap (lower–upper) before/after pruning, and (iii) failure mode taxonomy (e.g., timeout, solver infeasibility, loose pre-activation bounds). Wider validation across architectures/datasets is encouraged.

**Questions:**

(1) How does certification rate trade off with the RLT percentage when total time is capped? Do you have empirical heuristics for tuning $p$ (e.g., based on instability rate, Lipschitz proxies, or pre-activation interval widths)?

(2) Can you quantify the relaxation gap introduced by the pruning rules (Eqs. 28–31)? Under what conditions does this relaxation change the certificate (i.e., converts a would-be certificate into a failure)? Are these rules implicated in the observed failures?

(3) Extend Fig. 2 with $β$-CROWN-based pruning curves vs. class count to show how complexity and certified accuracy scale.

(4) How does the approach perform on CIFAR-10? Include the same ablations (RLT percentage, pruning on/off, baseline comparators, and $\epsilon$) for a like-for-like comparison.

Overall, widening the evaluation (architectures + baselines + $ϵ$) and adding the ablations above would substantively strengthen the paper’s empirical support and clarify where the proposed method is most beneficial.

---

### Official Review · Reviewer_QWud · 2025-10-26

**Soundness:** 2
**Presentation:** 2
**Contribution:** 3
**Rating:** 2
**Confidence:** 4

**Summary:**

While most neural network verifiers rely on linear relaxations of the networks to be verified, alternatives based on semidefinite relaxations provide tighter bounds, enabling more properties to be verified. However, Semidefinite Programming (SDP)-based verifiers usually need to solve one optimisation problem for each incorrect class, leading to high costs and slow verification. This work proposes an alternative formulation of the neural network verification problem as a quadratic program which verifies the robustness of a neural network across all classes simultaneously. The authors then relax this formulation to an SDP and introduce additional cuts based on McCormick envelopes. They further propose a method for pruning stably active neurons in a verification problem and, since the proposed method would break the chordal decomposition of the SDP, suggest a relaxation of the pruned problem which preserves the chordal structure. The proposed method is evaluated on a number of standard benchmarks and is shown to outperform previous verification approaches in terms of both runtime as well as the number of verified instances.

**Strengths:**

- Neural network verification is an important research topic
- The requirement to solve a number of separate optimisation problems is a major issue in SDP-based verification and tackling this is an important contribution.
- The experimental evaluation shows a significantly improved performance compared to the baselines that were evaluated

**Weaknesses:**

- To me, the biggest weakness of this work is the empirical evaluation. The results look impressive, but the authors use outdated baselines and do not compare against newer verification approaches. The settings used for e.g. $\beta$-CROWN are also questionable.
  - The experimental evaluation on $\beta$-CROWN seems unfair since the verifier is only run for very short time budgets (2-5 seconds) while the authors' proposed method is run for up to ~1600 seconds. Since $\beta$-CROWN is a complete verifier, a fair comparison would enable its branching and run it for the same time budget as $SDP_u$. The fact that the numbers reported here are not representative is also evidenced by comparing them to e.g. those reported by [6] where the gap between $\beta$-CROWN and the SDP verifiers is significantly smaller than the gap reported in this work.
  - $\beta$-CROWN has been improved by the introduction of general cutting planes generated by MILP solvers in [3] and branch-and-bound-inferred cutting planes in [4]. The method proposed by the authors should be evaluated against these newer works to benchmark how well it actually performs and not against the older $\beta$-CROWN. The authors conduct their experiments on very small models, I understand that this is standard in SDP-based verification so I do not hold this against them. However, it does seem quite likely that a MILP solver would scale to these sizes and would therefore be able to produce effective cutting planes in GCP-CROWN.
  - The authors do not compare their approach to newer SDP-based verifiers such as [6, 7, 8, 9], I know that at least for [8] the code has been made available so a comparison would be easy.
  - Given the small size of the networks which benchmarks are run on, I do wonder whether improved MILP verifiers or hybrid ones such as [2] would perform on these benchmarks
  - The paper's primary motivation is scaling to "large multi-class datasets", which also implies convolutional neural networks (CNNs). However, all experiments are conducted on small, fully-connected networks. The paper would be much stronger if it included at least a preliminary study on a small CNN.
- The related work section is missing multiple important references
  - The work cites early MILP verification approaches, but omits multiple later works on MILP-based verification which significantly improve upon the early (naive) approaches such as [1, 2]
  - All of the state-of-the-art work on incomplete verification is missing, including for example GCP-CROWN [3], BICCOS [4] and Marabou 2.0 [5]
  - A number of recent works on SDP-based verification are missing [6, 7, 8, 9]
- The removal of stably active neurons in a neural network verification context has previously been proposed by [10]. The approach proposed by the authors in this paper seems more general than the previously proposed method, but I think the previous method should still be cited. The previous method applies to fewer neurons but would not break the chordal sparsity pattern and would therefore not require the additional relaxation by constraints (28-31), so it would be interesting to see a performance comparison between the two
- The authors should provide some details on the networks that they train in Section 5.4 in terms of size. Being able to verify a neural network with 67 classes using SDP verification is great, but I wonder what the size and architecture of that network is.

### Minor weaknesses and typos
- Line 336: The authors say that "unmodeled quadratic terms appear in Constraint (5)". It should be made clearer that this is because of the removal of active neurons from the network which leads to new cross-layer dependencies, at the moment this is a bit difficult to understand.
- Line 108: Incomplete verifiers are derived into a wide variety of approaches --> Incomplete verifiers are **divided** into a wide variety of approaches
- Line 138: $W_K^j$, the $j^{th}$ row of matrix $W_K^j$ --> $W_K^j$, the $j^{th}$ row of matrix $W_K$
- Line 210: The triangular constraint (14) introduced in Ehlers (2017) tighten --> The triangular constraint (14) introduced in Ehlers (2017) tighten**s**
- Line 267: the DNN satisfy Property 2. --> the DNN satisf**ies** Property 2.
- Line 308: leverage the specific structure certification problem --> leverage the specific structure **of the** certification problem
- Line 357: where coefficient of $C_{k−1}$ are a linear combination --> where **the** coefficient**s** $C_{k−1}$ are a linear combination

### References
[1] Botoeva, E., Kouvaros, P., Kronqvist, J., Lomuscio, A. & Misener, R. (2020) Efficient Verification of ReLU-Based Neural Networks via Dependency Analysis. In: Proceedings of the AAAI Conference on Artificial Intelligence. 3 April 2020 pp. 3291–3299. doi:10.1609/aaai.v34i04.5729.

[2] Liao, Y., Genest, B., Meel, K. & Aryaman, S. (2025) Solution-aware vs global ReLU selection: partial MILP strikes back for DNN verification. doi:10.48550/arXiv.2507.23197.

[3] Zhang, H., Wang, S., Xu, K., Li, L., Li, B., Jana, S., Hsieh, C.-J. & Kolter, J.Z. (2022) General Cutting Planes for Bound-Propagation-Based Neural Network Verification. doi:10.48550/arXiv.2208.05740.

[4] Zhou, D., Brix, C., Hanasusanto, G.A. & Zhang, H. (2024) Scalable Neural Network Verification with Branch-and-bound Inferred Cutting Planes. Advances in Neural Information Processing Systems. 37, 29324–29353.

[5] Wu, H., Isac, O., Zeljić, A., Tagomori, T., Daggitt, M., Kokke, W., Refaeli, I., Amir, G., Julian, K., Bassan, S., Huang, P., Lahav, O., Wu, M., Zhang, M., Komendantskaya, E., Katz, G. & Barrett, C. (2024) Marabou 2.0: A Versatile Formal Analyzer of Neural Networks. In: Computer Aided Verification: 36th International Conference, CAV 2024, Montreal, QC, Canada, July 24–27, 2024, Proceedings, Part II. 25 July 2024 Berlin, Heidelberg, Springer-Verlag. pp. 249–264. doi:10.1007/978-3-031-65630-9_13.

[6] Lan, J., Brückner, B. & Lomuscio, A. (2023) A Semidefinite Relaxation Based Branch-and-Bound Method for Tight Neural Network Verification. Proceedings of the AAAI Conference on Artificial Intelligence. 37 (12), 14946–14954. doi:10.1609/aaai.v37i12.26745.

[7] Lan, J., Zheng, Y. & Lomuscio, A. (2023) Iteratively Enhanced Semidefinite Relaxations for Efficient Neural Network Verification. Proceedings of the AAAI Conference on Artificial Intelligence. 37 (12), 14937–14945. doi:10.1609/aaai.v37i12.26744.

[8] Chiu, H.-M. & Zhang, R.Y. (2023) Tight Certification of Adversarially Trained Neural Networks via Nonconvex Low-Rank Semidefinite Relaxations. In: Proceedings of the 40th International Conference on Machine Learning. 3 July 2023 PMLR. pp. 5631–5660. https://proceedings.mlr.press/v202/chiu23a.html.

[9] Ueda, R., Sato, T., Kobayashi, K. & Nakata, K. (2025) Interior-Point Vanishing Problem in Semidefinite Relaxations for Neural Network Verification. doi:10.48550/arXiv.2506.10269.

[10] Serra, T., Kumar, A. & Ramalingam, S. (2020) Lossless Compression of Deep Neural Networks. In: E. Hebrard & N. Musliu (eds.). Integration of Constraint Programming, Artificial Intelligence, and Operations Research. 2020 Cham, Springer International Publishing. pp. 417–430. doi:10.1007/978-3-030-58942-4_27.

**Questions:**

- How does the pruning method proposed by the authors compare to that proposed by Serra et al. in [10]?
- Why is $\beta$-CROWN only run for 2-5 seconds in the empirical evaluation?
- How does the method proposed by the authors compare to newer SDP-based as well as other NN verification works?
- What architecture is used for the self-trained models in Section 5.4?

---

> ### Author Response · Authors · 2025-11-20
>
> >*The experimental evaluation on $\beta$-CROWN seems unfair since the verifier is only run for very short time budgets (2-5 seconds) while the authors' proposed method is run for up to ~1600 seconds. Since  $\beta$-CROWN is a complete verifier, a fair comparison would enable its branching and run it for the same time budget as $SDP_u$. The fact that the numbers reported here are not representative is also evidenced by comparing them to e.g. those reported by [6] where the gap between  $\beta$-CROWN and the SDP verifiers is significantly smaller than the gap reported in this work.*
>
> We thank you for highlighting this important point. In our work, we propose an incomplete verifier *i.e.* we are only interested in computing a lower bound for the objective. Thus, we are comparing the performances against other incomplete verifiers. We then only reported the performances of $\beta$-CROWN used to calculate lower and upper bounds for each neuron. The running time of $\beta$-CROWN thus corresponds to the required time to compute those bounds.
>
>
> >*$\beta$-CROWN has been improved by the introduction of general cutting planes generated by MILP solvers in [3] and branch-and-bound-inferred cutting planes in [4]. The method proposed by the authors should be evaluated against these newer works to benchmark how well it actually performs and not against the older $\beta$-CROWN. The authors conduct their experiments on very small models, I understand that this is standard in SDP-based verification so I do not hold this against them. However, it does seem quite likely that a MILP solver would scale to these sizes and would therefore be able to produce effective cutting planes in GCP-CROWN.*
>
> We already discussed GCP-CROWN in our extended related work (see Appendix D.3). Indeed we are mostly interested in improving and scaling SDP based verifier. We, thus, only compare our new approach with SDP-based methods and other well known incomplete verifiers.
>
> >*The authors do not compare their approach to newer SDP-based verifiers such as [6, 7, 8, 9], I know that at least for [8] the code has been made available so a comparison would be easy.*
>
> The methods proposed in [6,7,9] are also complete verifiers and, to the best of our knowledge, their code has not been published.
> Regarding the method in [8], we found an undocumented Matlab library that we were unable to run and might not be trivially compatible with most deeplearning python libraries.
>
> >*Given the small size of the networks which benchmarks are run on, I do wonder whether improved MILP verifiers or hybrid ones such as [2] would perform on these benchmarks*
>
> While interesting, the suggested reference was published late after ICLR submission deadline (10/26/25), so we did not compare ourselves to it.
> We did, however, test submitting (QP) to the commercial solver Gurobi, and the results showed that directly solving the quadratic formulation is too slow and does not scale even for the smallest network considered in our experiences : 6x100 (timeout = 3h).
>
> >*The paper's primary motivation is scaling to "large multi-class datasets", which also implies convolutional neural networks (CNNs). However, all experiments are conducted on small, fully-connected networks. The paper would be much stronger if it included at least a preliminary study on a small CNN.*
>
> To the best of our knowledge no SDP relaxation is able to scale with ResNet and large CNN. This would imply scaling SDP matrices to a large number of parameters by taking into account convolutional kernel induced sparsity, to deal with residual connections and batch-norm. While this is indeed a very exciting line of work, it is orthogonal to our contributions.

---

> ### Author Response · Authors · 2025-11-25
>
> >*The related work section is missing multiple important references
> The work cites early MILP verification approaches, but omits multiple later works on MILP-based verification which significantly improve upon the early (naive) approaches such as [1, 2]
> All of the state-of-the-art work on incomplete verification is missing, including for example GCP-CROWN [3], BICCOS [4] and Marabou 2.0 [5]
> A number of recent works on SDP-based verification are missing [6, 7, 8, 9]*
>
> Thank you for pointing out these references to us. We will cite them in the state of the art section of our final version.
>
> >*The removal of stably active neurons in a neural network verification context has previously been proposed by [10]. The approach proposed by the authors in this paper seems more general than the previously proposed method, but I think the previous method should still be cited. The previous method applies to fewer neurons but would not break the chordal sparsity pattern and would therefore not require the additional relaxation by constraints (28-31), so it would be interesting to see a performance comparison between the two*
>
> Thank you for the reference, we will add the reference to the state of the art. As stated in your review, we agree that our approach is a generalization of the one proposed in [10]. However, to the best of our knowledge, their code has not been published, this is why we do not present a comparison with their approach.
>
> >*The authors should provide some details on the networks that they train in Section 5.4 in terms of size. Being able to verify a neural network with 67 classes using SDP verification is great, but I wonder what the size and architecture of that network is.*
>
> All details of the networks we train are reported in section C of the appendices, we will add a pointer in section 5.4 to the appendix.
> The network is a feed forward network of sizes 784x100 and 100-N with N being the total number of classes : $N \in [10,67]$. We trained it with during 100 epochs and obtain a final accuracy score of 75.2%
>
> **Minor comments**
>
> >*Line 336: The authors say that "unmodeled quadratic terms appear in Constraint (5)". It should be made clearer that this is because of the removal of active neurons from the network which leads to new cross-layer dependencies, at the moment this is a bit difficult to understand.*
>
> We thank you for your careful reading of the manuscript. We will correct the mentioned typos and will rewrite the sentence “However, unmodeled quadratic terms appear in Constraint (5), i.e..” in the final version as :
> “The removal of active neurons from the network leads to new cross-layer dependencies that are modeled by quadratic terms not occurring in the chordal decomposition. These unmodeled quadratic terms appear in Constraint (5), i.e. …”
>
> The other typos have been corrected in the revised manuscript

---

> > ### Comment · Reviewer_QWud · 2025-11-25
> >
> > I'd like to thank the authors for taking the time to respond to my review. Regarding the points that were raised:
> >
> > > In our work, we propose an incomplete verifier i.e. we are only interested in computing a lower bound for the objective. Thus, we are comparing the performances against other incomplete verifiers.
> > I understand that the proposed method is an incomplete verifier, however, when running e.g. GCP-CROWN with a set timeout, it is technically also an incomplete verifier (since, if time is insufficient, the verifier will not return a definitive result as to whether a given instance is robust). Even leaving this aside, in my opinion, it makes sense to compare different verifiers using the same time budget, e.g. enabling branching in bound-propagation-based verifiers. Other works also use similar timeouts for different verifiers to enable a fair comparison.
> >
> > More generally, I would argue that even though SDP-based verifiers are incomplete, one "bounding step" with them is significantly more expensive than obtaining bounds with a linear-bound-based method such as GCP-CROWN. I would therefore disagree with the authors' idea of comparing SDP-based verification with single bound propagation steps in competing tools. The trade-off here is whether a given time budget should be spent on one bounding step with more precise relaxations (SDP) or whether it should be spent on many bounding steps employing much cheaper, but less precise relaxations (linear bound propagation such as GCP-CROWN).
> >
> > This also ties in with the comments made regarding CNNs - I agree that no SDP verifier can handle these so far, but bound-propagation-based verifiers have no issues dealing with CNNs. Given their popularity in deep learning, not being able to verify the robustness for convolutional models is a weakness compared to existing approaches.
> >
> > I'd further like to thank the authors for fixing the typos and for the revisions of the manuscript based on the points that I raised, this is much appreciated. I will take all these points into account and reevaluate my score based on them.

---

> > > ### Author Response · Authors · 2025-11-27
> > >
> > > We thank the reviewer for the thoughtful comments. We posted the revised version of the manuscript, with modifications highlighted in blue.
> > >
> > > Our main contribution is a model agnostic to the number classes which is a first step toward scaling certification on medium to large size datasets. We agree that SDP-based relaxations have high computational cost, but lead to very tight bounds. To tackle this computational challenge, we propose in this paper to exploit the sparsity of neural networks computation graphs by combining i) chordal decomposition with ii) stable neurons pruning.
> > > Our new SDP formulation is a first step towards tackling modern architectures such as CNN. A future work will be to reduce the CPU time of SDP resolution by for instance designing a sub-gradient algorithm within a Lagrangian duality framework (Dathathri et al, 2020). Moreover, complete certification could be handled by developing a B&B based on quadratic convex relaxations that reaches the value of the SDP bound (see Billionnet et al, 2016).
> > >
> > >
> > >
> > >
> > > (Billionnet et al, 2016) Billionnet A, Elloumi S, Lambert A,“Exact quadratic convex reformulations of mixed-integer quadratically constrained problems.” Mathematical Programming 158(1):235–266 (2016), ISSN 1436-4646, URL http://dx.doi.org/10.1007/s10107-015-0921-2.
> > >
> > > (Dathari et al, 2020) Dathathri, Sumanth, et al. "Enabling certification of verification-agnostic networks via memory-efficient semidefinite programming." Advances in Neural Information Processing Systems 33 (2020) https://arxiv.org/pdf/2010.11645

---

### Official Review · Reviewer_tKZa · 2025-10-30

**Soundness:** 2
**Presentation:** 2
**Contribution:** 3
**Rating:** 6
**Confidence:** 4

**Summary:**

Paper develops an SDP relaxation of neural network verification methods intended to verify a large number of classification/prediction criteria by considering the criteria holistically as a single problem rather than individual problems.

**Strengths:**

1.	Advances the state of practice in relaxation-based approaches for NN verification based on SDP relaxations.
2.	Increases the scale of NN’s were SDP relaxations can be used.

**Weaknesses:**

1.	The literature review is generally restricted to a single paper from 2022 and the rest being older than that.  There is quite a bit of more recent work, esp. https://arxiv.org/abs/2506.06665, which has very much improved the state of the art of relaxations to SDP for NN verification.  Oddly, this paper appears in the reference list, but is not referred to in the text. So, I am a bit concerned that this and other recent work is not compared against.
2.	Some parts of the model are unclear.  For example, it is not clear whether or not the beta (binary) variables are relaxed or not.  From the notation, I think the beta variables remain binary.  However, my understanding is that Mosek does not support mixed integer SDPs.  Either way, I have a couple questions below.

**Questions:**

1.	Are the binary variables relaxed or binary in the final relaxation?
a.	If relaxed, I would expect the relaxation to be weaker than the enumeration of all classes using SDP with the cuts that are specific to each class.  (because the optimization can assign fractional values to beta).  Is this the case? Do you have CPU and solution quality comparisons between your model enumerated per class vs. your one large model?  I am not completely sure if SDP_t is exactly this or not.
b.	If not relaxed, is the main value of the combined model that it can take advantage of the branching strategies and pruning capabilities of the mixed integer solver, and not have to do a complete enumeration of all classes (e.g., the search tree is essentially one layer below the root node, with each leaf essentially corresponding to one verification class and many leaves pruned and never evaluated)?

---

> ### Author Response · Authors · 2025-11-20
>
> >*The literature review is generally restricted to a single paper from 2022 and the rest being older than that. There is quite a bit of more recent work, esp. https://arxiv.org/abs/2506.06665, which has very much improved the state of the art of relaxations to SDP for NN verification.*
>
> We would like to note that this reference appears in our bibliography as we refer to it in our extended related work (see Supplementary Material, Section D.1), where we provide a more extensive overview of the state of the art of SDP-based certification methods. We did not compare our approach with the SDP-CROWN method since SDP-CROWN was developed in the context of $\ell_2$​-norm adversaries while in this paper we focus on $\ell_\infty$​-norm adversaries.
>
>
> >*Some parts of the model are unclear. For example, it is not clear whether or not the beta (binary) variables are relaxed or not. From the notation, I think the beta variables remain binary. However, my understanding is that Mosek does not support mixed integer SDPs. Either way, I have a couple questions below.*
>
> Thank you for raising this point. We would like to clarify that our SDP relaxation does include the relaxation of binary variables. This is induced by constraint (21) (i.e., z² = z). Therefore, the numerical results reported do not include a B&B on the binary variables. Note that the introduction of RLT cuts significantly reinforces this relaxation. Our approach extends directly to solving SDPs with binary variables. Since no SDP solver is designed to solve SDPs with binary variables, implementing a dedicated B&B could be interesting for future work. While providing a better quality bound, this would also greatly increase the computation time, thus penalizing scaling.
>
> ### Questions
>
> >*Are the binary variables relaxed or binary in the final relaxation?*
>
> Yes, the binary variables are relaxed in (SDP).
>
> >*If relaxed, I would expect the relaxation to be weaker than the enumeration of all classes using SDP with the cuts that are specific to each class. (because the optimization can assign fractional values to beta).*
>
> You are absolutely right, if we were to compare our method with a targeted formulation computed separately for each class and without any RLT cuts, our approach would likely appear weaker.
> However, this relationship between the optimal values of the two problems is no longer true as we are able to incorporate substantially more RLT cuts in ($SDP_u$) than in ($SDP_t$). The values of the bounds are then no longer comparable.
>
>
> >*Is this the case? Do you have CPU and solution quality comparisons between your model enumerated per class vs. your one large model? I am not completely sure if $SDP_t$ is exactly this or not.*
>
> In terms of solution quality for models with RLT, we observe in the numerical results that the bounds obtained with ($SDP_u$) are tighter than those obtained with ($SDP_t$), since the number of data certified by ($SDP_u$) is significantly greater than that certified by ($SDP_t$).
> In terms of CPU time, since the two models have approximately the same size, solving ($SDP_u$) or ($SDP_t$) on one target requires a similar computing time. Thus, our new approach is significantly faster since it requires only a single resolution (that of ($SDP_u$)) to certify a data point, whereas with targeted approaches, it is necessary to resolve the model ($SDP_t$) as many times as there are classes in the network. The computation time of our new approach is therefore reduced by a factor equal to the number of classes in the network.
> It should also be noted that this single resolution allows us to add a very large number of RLT cuts in (SDP) and to obtain (experimentally) a bound that is very often tighter than that obtained by the targeted method.
>
> >*If not relaxed, is the main value of the combined model that it can take advantage of the branching strategies and pruning capabilities of the mixed integer solver, and not have to do a complete enumeration of all classes (e.g., the search tree is essentially one layer below the root node, with each leaf essentially corresponding to one verification class and many leaves pruned and never evaluated)?*
>
> Yes. Developing a B&B based on the bound obtained with ($SDP_u$) to calculate the solution in binary variables would further improve the value of the bound. However, there is no mixed-SDP solver, and this would require us to develop our own B&B, which would not benefit from the highly effective heuristics developed for binary variable optimization that have been introduced in commercial solvers over the last 20 years. This extension is a future work.
> It is important to note that applying this B&B could significantly increase computation time, and a good balance will indeed need to be reached between improving the bound and scaling.
> We did, however, test submitting (QP) to the commercial solver Gurobi, and the results showed that directly solving the quadratic formulation is too slow and does not scale.

---

### Author Response · Authors · 2025-11-20

We thank the reviewers for their careful reading of our manuscript and their insightful feedback. We have provided detailed responses to each comment. Still, we would like to highlight in particular the following points.

Several observations addressed comparisons with complete branch-and-bound verifiers such as $\beta$-CROWN based Branch and Bound, or Marabou. Our work introduces an incomplete verifier, so our primary comparisons focus on methods aimed at producing lower bounds on the certification objective.

Concerning the experimental set-up, to the best of our knowledge, SDP-based methods have not yet been deployed on modern CNN architectures such as ResNets, given the challenge of scaling to millions of parameters and handling residual connections and normalization layers. Although advancing the scalability of SDP-based models is an important and exciting research direction, it remains orthogonal to our current contributions.

---

> ### Author Response · Authors · 2025-11-27
>
> We thank the reviewers for the thoughtful comments. We would like to inform you that the revised version of the manuscript has been posted, with all modifications highlighted in blue.

---

> > ### Author Response · Authors · 2025-12-02
> >
> > We sincerely thank the reviewers for their insightful comments and their responsiveness during this **short** discussion period. We have carefully addressed the requests and conducted additional experiments that further confirm the relevance and robustness of our proposed method.
> >
> > Our paper introduces a novel multiclass quadratic formulation for modeling incomplete certification of deep neural networks. As all reviewers emphasized, our work tackles a critical limitation of SDP-based verification methods: their inability to scale to large datasets. This is a key challenge in neural network verification, and our contributions represent a significant step forward. We provide both substantial theoretical insights and practical innovations that directly address this limitation:
> > a model agnostic to the number classes, which is a first step toward scaling certification on medium to large size datasets.
> > SDP-based relaxations have high computational cost, but lead to very tight bounds. To tackle this computational challenge, we propose in this paper to exploit the sparsity of neural networks computation graphs by combining chordal decomposition with stable neurons pruning. Our new SDP formulation is a first step towards tackling modern architectures.
> >
> > We sum up hereafter the main points discussed during this rebuttal. In our work, we propose an incomplete verifier i.e., we are only interested in computing a lower bound for the objective, which justifies the comparison with incomplete verifiers only. Moreover, we ran a large set of additional experiments to further assess the performance of our approach. In particular, we added the following ablations:
> >
> > (i) certified fraction vs. number of unstable ReLUs: CPU time is always reduced while the certification accuracy may even be improved.
> >
> > (ii) average bound gap (lower–upper) before/after pruning: bound values remain nearly unchanged, confirming robustness to pruning of active neurons.
> >
> > (iii) certified performance/percentage of RLT: the CPU time does not blow up,  while accuracy improves significantly with 100% of RLT.
> >
> > (iv) impact of the pruning on large sized networks (model size/bound): as expected trend, the larger the network and the perturbation, the lower is the number of constraints when full pruning is applied. These results demonstrate that our approach outperforms existing SDP-based incomplete verifiers in both efficiency and accuracy, while addressing a fundamental scalability bottleneck.
> >
> > We believe these contributions and experimental validations clearly establish the significance of our work and its potential impact on advancing neural network verification.

---

### Meta-Review · Area_Chair_RALh · 2026-01-06

**Summary:**

While the reviewers acknowledge the relevance of the problem tackled in this work, they express concerns related to the discussion of and comparison with the existing literature (in particular recent work), the clarity of some aspects of the methodology, the influence of hyper-parameter p, the applicability of the method beyond fully-connected networks, and some specific aspects of the methodology and  evaluation.

**Reviewer Concerns:**

The specific methodological and evaluation questions raised by the reviewers were convincingly addressed by the authors. However, the AC expects that the question of discussion of and comparison with the recent literature, raised by most reviewers, would not have fully convinced the reviewers, as also suggested by the follow-up comment of Reviewer QWud.

**Reviewer Scores:**

Two reviewers participated in the discussion but without clearly expressing their intention to raise their score, although Reviewer QWud mentioned that they would reevaluate their score based on the feedback. Because the shared concern of comparison with the recent literature was not fully addressed in the rebuttal, the AC expects that most reviewers would not have increased their score. The AC therefore recommends the authors to revise their paper based on the reviewers' comments, with a focus on strengthening the empirical evaluation, and resubmit it to a future venue.

---

### Decision · Program_Chairs · 2026-01-26

Reject